Resource

# AUTOENCODIX: a generalized and versatile framework to train and evaluate autoencoders for biological representation learning and beyond

Maximilian Josef Joas [1] ✉, Neringa Jurenaite [2], Dušan Praščević[1],
Nico Scherf [1,3] & Jan Ewald [1] ✉

In recent years, autoencoders, a family of deep learning-based methods for representation learning, are advancing data-driven research owing to their variability and nonlinear power for multimodal data integration. Despite their success, current implementations lack standardization, versatility, comparability and generalizability. Here we present AUTOENCODIX, an open-source framework, designed as a standardized and flexible pipeline for preprocessing, training and evaluation of autoencoder architectures. These architectures, such as ontology-based and cross-modal autoencoders, provide key advantages over traditional methods by offering explainability of embeddings or the ability to translate across data modalities. We apply the method to datasets from pan-cancer studies (The Cancer Genome Atlas) and single-cell sequencing as well as in combination with imaging. Our studies provide important user-centric insights and recommendations to navigate through architectures, hyperparameters and important tradeoffs in representation learning. These include the reconstruction capability of input data, the quality of embedding for downstream machine learning models and the reliability of ontology-based embeddings for explainability.

The key to gaining insights into complex systems, such as cells and organisms, lies in accurate feature representation of high-dimensional, multimodal data. This is critical with the rise of faster and cheaper large-scale data generation. However, multimodal high-throughput data, such as molecular and genetic (multi-omics) datasets, suffer from the curse of dimensionality, as the number of features (such as genes or molecules) often exceeds the sample size (patients or cells)[1,2]. Dimensionality reduction and unsupervised learning are thus essential for uncovering patterns, identifying biomarkers, enabling efficient

supervised learning and robustly defining patient similarities, a key to personalized medicine.

Deep-learning-based autoencoders (AEs) have recently become popular for representation learning and unsupervised dimension reduction. They employ an encoder–decoder structure with a low-dimensional bottleneck (latent space)[3]. Compared with principal component analysis (PCA) or uniform manifold approximation and projection (UMAP), AEs offer advantages such as nonlinear embeddings, flexible multimodal data integration and synthetic data

[1]Center for Scalable Data Analytics and Artificial Intelligence (ScaDS.AI) Dresden/Leipzig, Leipzig University, Leipzig, Germany. [2]Center for Scalable Data Analytics and Artificial Intelligence (ScaDS.AI) Dresden/Leipzig, Technical University Dresden, Dresden, Germany. [3]Max Planck Institute for Human Cognitive and Brain Sciences, Leipzig, Germany. ✉e-mail: maximilian.joas@uni-leipzig.de; jan.ewald@uni-leipzig.de

**Fig. 1 | A schematic illustration of the open-source framework.** AUTOENCODIX provides a full pipeline for training and evaluation of various AE architectures. Upon release, five common or emerging AE architectures are available and shown as sketches. See 'Framework and AE implementation details' in Methods and 'Design and features' section for a detailed description. *t*-distributed stochastic neighbor embedding (*t*-SNE).

generation[3,4]. Advanced AE variants even enable modality translation (cross-modal AEs)[5] or incorporate domain knowledge for explainable latent spaces[6–8].

These strengths make AEs particularly valuable for integrating multimodal biological data, reflecting the complex interactions of genetic and molecular factors[9–15]. Single omics measurements shed light on one layer of these interactions, such as gene expression (transcriptomic), but cannot fully capture the disease causes, which perturb cells across all layers including epigenetic, genomic, metabolomic or proteomic states[16]. Hence, a dataset such as The Cancer Genome Atlas (TCGA) is valuable as it contains multi-omic measurements of patients across major cancer types[17]. However, developing methods to integrate data modalities and shed light on the complex interplay via low-dimensional representations is still a challenge[18].

Despite active research, broad application of AEs is hindered by the lack of standardized, user-friendly frameworks. No single framework currently combines key aspects such as state-of-the-art AE variants, multimodal integration, flexible configuration and hyperparameter optimization. The comparable system Rapidae[19] addresses some needs but focuses on time-series data, which are rare in molecular studies, and lacks a full pipeline approach.

To address this gap, we developed AUTOENCODIX, an open-source, PyTorch-based[20] framework unifying these aspects for biological representation learning with AEs. While designed for large-scale biomolecular data, AUTOENCODIX's flexible architecture supports other fields as well. By offering unified AE implementations and simple parameterization, we provide a broad benchmark of AE variants for tasks such as embedding performance in prognostic models, modality translation and ontology-based explainability.

## Results

### Design and features

Our Python- and PyTorch-based framework, AUTOENCODIX, was developed to broaden the applicability of AEs for multi-omics data integration and beyond (Fig. 1). AUTOENCODIX addresses limitations of current frameworks by: (1) supporting multiple AE architectures, (2) offering an end-to-end pipeline from preprocessing to downstream analyses (visualization and embedding evaluation), (3) enabling multimodal integration, (4) including hyperparameter tuning, (5) ensuring flexible and explainable usability and (6) providing easy extensibility.

AUTOENCODIX accepts any number of data modalities, including annotations, as data frames. While we offer scaling of input data modalities, we assume that each modality is normalized beforehand for technical biases. For single-cell data integration, we include preprocessing steps for the AnnData h5ad-files[21]. Input and options are defined via YAML configuration files, enabling transparent, reproducible analyses and easy benchmarking while maintaining control.

All pipeline steps can be run individually or in a single command including: preprocessing, model training, visualization and evaluation of embeddings on user-defined machine learning (ML) tasks (for instance, subtype classification or survival prediction). A detailed description of the preprocessing options and hyperparameters is provided in the AUTOENCODIX documentation (https://github.com/jan-forest/autoencodix/blob/main/ConfigParams.md).

At its core, AUTOENCODIX implements five AE architectures capturing recent advances. Two baseline models are included: a vanilla AE (Vanillix) and a variational autoencoder (VAE, named Varix), both with fully connected encoder and decoder layers and a bottleneck. Layers include linear, batch normalization, dropout and rectified linear unit (ReLU) activations. A stacked VAE (Stackix), also known as a hierarchical AE, extends this design for multi-omics integration with good performance[22]. It does so by training independent VAEs for each modality, then concatenating embeddings for a stacked VAE, which reduces the sensitivity to differences in feature scale or distributions across modalities.

To incorporate biological knowledge and offer explainability, AUTOENCODIX includes ontology-based AEs. Our Ontix model is distilled from previous models[6–8] and integrates a user-defined ontology such as pathways directly into the decoder, aligning latent dimensions with interpretable biological processes. Finally, we implemented a

cross-modal VAE (X-modalix) inspired by Yang et al.[23], which enables translation between modalities, including image data. Our architectures rely on paired measurements that are difficult to obtain in multi-omics, and unpaired integration remains a challenge[24]. However, our pipeline can be used for both vertical and horizontal integration when users transpose feature and sample space. For future versions of AUTOENCODIX, current architectures will serve as building blocks of new developments such as VAEs for mosaic integration[25], uncoupled VAE with regularized inverse optimal transport[26] or masked VAEs for imputation.

A distinguishing feature of AUTOENCODIX is its emphasis on explainability and sustainability. While explainable AEs remain an active research area, Ontix provides interpretability, linking latent dimensions to biological pathways. Beyond technical capabilities, we commit to long-term maintenance, continuous development to include emerging trends and community-driven contributions via a contribution guide. This contrasts with existing implementations that lack support, hindering adoption.

Another key limitation of other frameworks is the absence of hyperparameter tuning, visualization and embedding evaluation. AUTOENCODIX integrates these directly, since tuning is essential for real-world applications in biomedical representation learning. Users can explore embeddings through multiple visualizations and quantitatively evaluate performance on predictive tasks, closing gaps left by previous implementations.

### Navigating the zoo of AEs for representation learning

**Tradeoffs behind $\beta$-VAE in reconstruction.** VAEs are probabilistic generative latent models, where the latent space is modeled as a (typically multivariate normal) distribution. To train, the encoder approximates this latent (posterior) distribution and the loss function is extended by a similarity measure of the latent distribution with the prior using Kullback–Leibler divergence (KL loss). Hence, they are trained for a tradeoff between minimizing the reconstruction error (mean-squared error (MSE)), and the compactness of the latent space. The tradeoff is captured by the weighting $\beta$ between the loss terms (see 'Framework and AE implementation details' section in Methods), and we analyzed the impact of $\beta$ by training five VAEs with different $\beta$ values from zero to ten, which are annealed over 1,000 epochs. Since all architectures, besides Vanillix, are VAEs and $\beta$ is a critical hyperparameter and not tunable via loss function optimization, its configuration is a necessary preceding step.

By increasing $\beta$, we can analyze the impact of the KL loss on the reconstruction capability, the latent space density (the average coverage per latent dimension) and the total correlation of latent dimension, a measure of the independence of latent dimensions (disentanglement, compare with PCA)[27]. The coverage calculation compares to perfectly uniformly distributed samples in each dimension (1 for uniform coverage, 0 for single point distribution coverage; see 'Experiments' section in Methods). The total correlation is a nonnegative value and a quantification of the shared information between variables (latent dimensions), where a value of zero represents complete dependence (see 'Experiments' section in Methods). The reconstruction capability, expressed as the explained variance $R^2$, is the highest for very small $\beta = 0.01$ (Fig. 2b). This behavior is desirable when the low-dimensional representation must contain the maximum amount of information. However, larger weights of KL loss result in latent dimensions with higher density and slightly higher disentanglement (lower total correlation), which is advantageous to obtain disentangled and interpretable dimensions. Additionally, when using VAEs to generate synthetic data, a dense latent space is preferable to generate meaningful synthetic data from any point in the latent space.

On the basis of these results, we use the three $\beta$ values (0.01, 0.1 and 1) with reasonable tradeoffs for further testing.

**Architecture comparison and hyperparameter analysis with AUTOENCODIX.** A main motivation for AUTOENCODIX is the ability to compare different types of AEs and hyperparameter configurations to identify the best one for a given dataset or task. Although there have been some benchmarks[11,22], there are little to no recommendations regarding the strengths and weaknesses of architectures, the impact of tuning or the performance of embeddings in downstream ML tasks.

As a first step, we analyzed four types of AE on two datasets (TCGA and single-cell) with up to three different data modalities and three latent dimensions (Fig. 2a and 'Experiments' section in Methods). The three levels of dimensionality represent typical use cases: a two-dimensional visualization ($L_{dim} = 2$), an intermediate dimensionality ($L_{dim} = 8$) and one associated to the number of Reactome top-level pathways representing the latent space for Ontix ($L_{dim} = 29$). In detail, an Ontix with $L_{dim} = 29$ has a one-to-one association of pathways and node connections in its two-layered sparse decoder (see 'Framework and AE implementation details' section in Methods). Lower dimensional embeddings ($L_{dim} = 2$ and $L_{dim} = 8$) with Ontix, mainly for benchmark completeness, are realized by the introduction of a third fully connected layer in both the decoder and encoder. Further, using in-built Optuna support[28], we tuned the hyperparameters for the learning rate, weight decay, hidden layer structure and dropout rate. To show the effect of tuning, we also determined the performance of the respective standard AE configuration.

AEs are used to determine low-dimensional embeddings enabling better training of supervised learning models, which is not captured by the loss optimization. Hence, we consider two criteria: the reconstruction capability and the reconstruction loss (see Fig. 3a and compare to Supplementary Fig. 1 for nonaggregated results) and the embedding performance on downstream regression and classification tasks using three ML algorithms (see Fig. 3b for the performance in linear ML and compare to Supplementary Fig. 2 for all three). Tasks are based on sample annotations (see the list in Supplementary Table 3) and include subtype classification or prediction of survival (regression). As baseline, we determined the embedding performance as the normalized improvement over training on randomly selected features (matching the number of latent dimensions). We also compared the embeddings with PCA and UMAP.

We observe that the reconstruction capability of Vanillix or VAEs with low $\beta$ is more precise (Fig. 3a). Similarly, the smaller the latent space, the lower the reconstruction capability, where, in particular, two-dimensional VAEs struggle to precisely recover high-dimensional data. Both the results and the general tradeoffs in reconstruction capability are largely independent of data modality, dataset and VAE type.

The stacked VAE (Stackix, see also 'Framework and AE implementation details' section in Methods) demonstrates similar performance but is not directly comparable to other architectures because of its uncoupled two-phase training process for each VAE and that the final reconstruction loss is calculated with embeddings as input from each data modality, rather than on features as in other types of AE. Figure 3a shows the explained variance $R^2$ of the reconstructed individual VAE embeddings per data modality on which the final stacked VAE is trained. Interestingly, a lower $\beta$ configuration has a higher impact in Stackix and is crucial to obtain valuable embeddings. Notably, the ontology-based VAE (Ontix, see also 'Framework and AE implementation details' section in Methods) demonstrates diminished reconstruction ability, which is justified by the constraints of its sparse decoder reflecting the gene involvements in Reactome pathways.

Compared to nontuned AEs, the reconstruction performance improves primarily for two-dimensional vanilla AE and VAEs and for the stacked VAE with low $\beta$ (Fig. 3c). On the other end, ontology-based VAEs with latent dimension of 8 and 29 benefit the most from tuning, while other architectures are comparable in performance when using standard configurations.

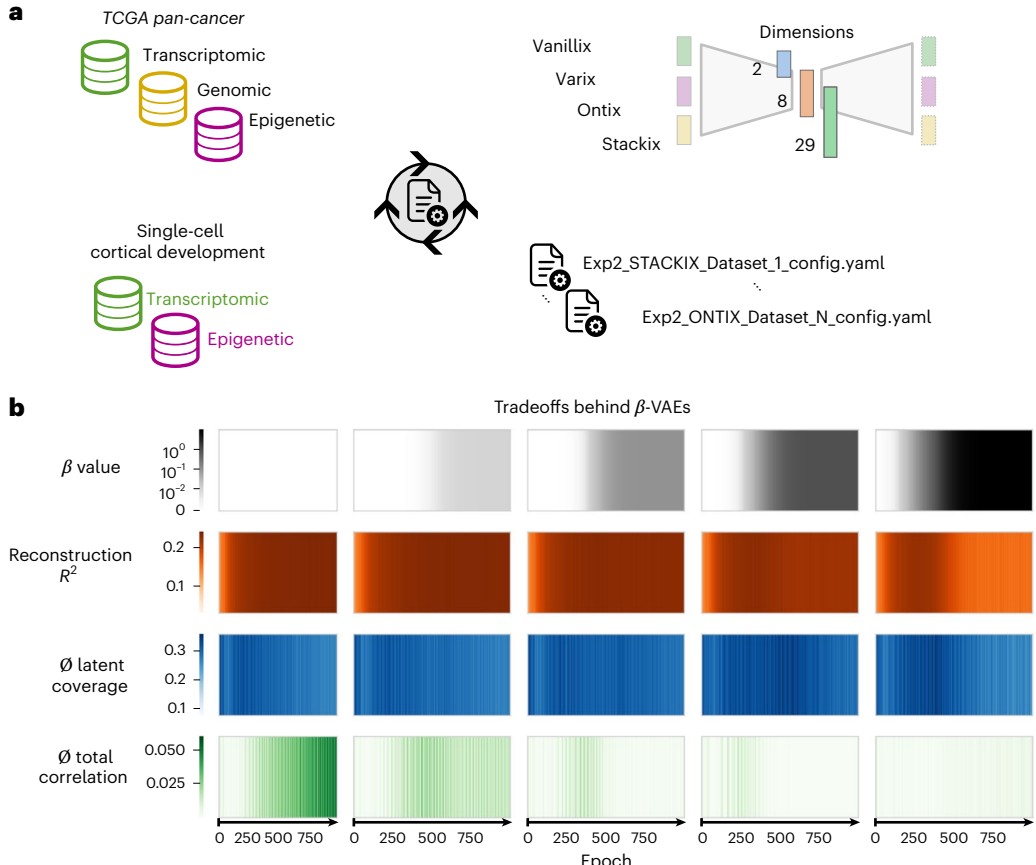

**Fig. 2 | AUTOENCODIX and its architectures for representation learning. a**, An overview of the datasets used, their Omics modalities and the AE architectures for performance comparison. **b**, An illustration of the tradeoffs behind the $\beta$ configuration in VAEs via training with annealing of the two-dimensional Varix AE on the TCGA dataset and all the data modalities with five different $\beta$ values. $\varnothing$ represents the mean overall latent dimensions.

**Reconstruction capability versus embedding quality.** The performance of embeddings for prognostic models is crucial for applications. In our experiments across two datasets, all data modality combinations and three ML algorithms, no architecture consistently outperforms the others and results vary depending on the combination of architecture and algorithm (see Fig. 3b, using linear ML algorithms, and compare to Supplementary Fig. 2). Including other methods (PCA and UMAP) and using random features as a reference, we show that all methods outperform random feature selection. Further, we see that linear ML algorithms benefit in performance over randomly selected features from higher dimensions, but RandomForest and support vector machines (SVM with radial kernel) show no clear trend or even inverse behavior in the case of UMAP (Supplementary Fig. 2). From the user perspective, we conclude that the evaluation of the embedding for downstream tasks is crucial and all architectures should be compared in practice, since there is no general superiority. This is, in particular, true for VAE with higher $\beta$ values or ontology-based VAEs, where we can see that worse reconstruction capability does not correlate with embedding quality. The same applies when analyzing the impact of tuning on embedding quality. Except for ontology-based VAEs with latent dimension of 8 and 29, the embeddings show no consistent improvement (Fig. 3d). This underscores that improvements in reconstruction do not necessarily result in better embeddings. However, large improvements in reconstruction, as for some ontology-based VAEs and others, will probably improve embeddings for prognostic model training. Ontix is interesting here since it is the only architecture showing improvements in reconstruction capabilities through hyperparameter optimization (Fig. 3c), which translate into better embeddings (Fig. 3d).

On the other hand, our standard configurations of AEs are fairly robust for applications without tuning. An exception is the ontology-based VAE, which has, in general, different hyperparameter requirements. For example, they show a necessity for lower dropout rates and faster learning rates (Supplementary Fig. 3).

## Ontix for explainability of latent space
**The importance of ontology for gaining biological insights.** A key advantage of AEs over methods such as UMAP is their deterministic embedding generated by the encoder structure. Importantly, they allow greater explainability of latent dimensions by incorporating biological knowledge. This is achieved by restricting the decoder to feature connectivity according to an ontology (Fig. 4a), creating explainability 'by design' rather than relying on post hoc feature importance. In our experiments, the decoder reflects a two-level ontology hierarchy and supports overlapping gene pathway memberships by restricting weights only where no ontology connection exists (see 'Framework and AE implementation details' in Methods).

We tested this on TCGA (Fig. 4) and single-cell cortex data (Supplementary Fig. 4) with two ontology types: chromosomal gene location and biological pathways (Reactome[29]). Chromosomal location is ideal for benchmarking since there is no overlap between features and ontology, and sex chromosomes provide a testable expectation of signal. Indeed, TCGA data show a strong signal on the X chromosome (Fig. 4b), though none on Y owing to its small size and few genes. Without additional loss weighting, some ontology dimensions are disregarded in favor of major variance.

A practical use of explainable latent dimensions is shown in Fig. 4d, where latent intensities reflect pathway activity by tissue of

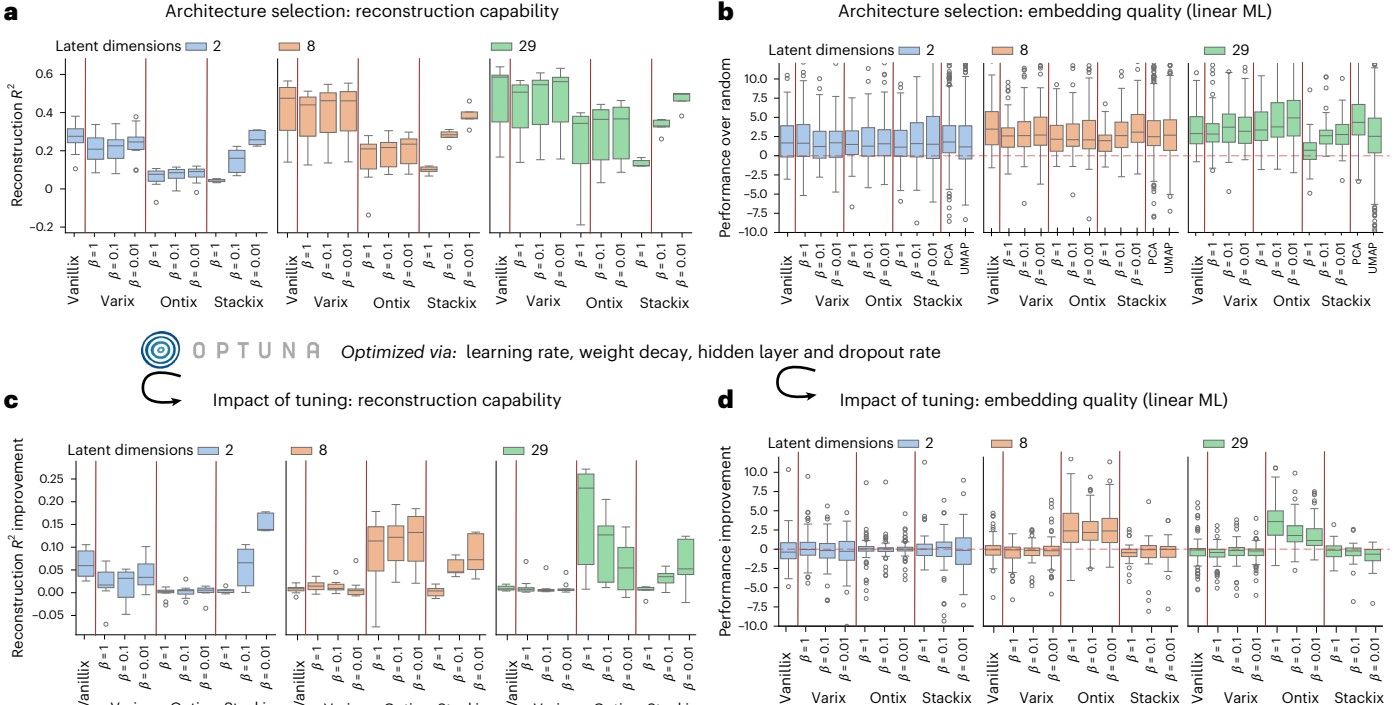

**Fig. 3 | Benchmarking of architectures. a,c**, The aggregated reconstruction capability, defined as the explained variance $R^2$ of the input versus reconstruction, of all the combinations ($n = 10$) as shown in Fig. 2a of fully tuned AEs (**a**) and in comparison to nontuned AEs ($R^2_{tuned} - R^2_{nontuned}$, interconnection illustrated by the curved arrows) (**c**). **b**, The aggregated results for the embedding performance of each approach on multiple downstream ML tasks (via linear regression or logistic regression, $n = 86$), shown as the normalized performance over randomly selected features with a similar dimension to the AE. **d**, The embedding performance in comparison to nontuned AEs. Dashed red lines in **b**,**d** indicate the boundary (zero) between improvement (positive values) and worsening of the performance (negative values). Box plots show the median, interquartile range, whiskers extending to 1.5× the interquartile range and outliers as points.

cancer origin. For example, central nervous system cancers appear distinct, linked to immune processes, vesicle transport and neuronal system activity.

Beyond explainability, Ontix shows strong performance in downstream tasks using Reactome-based embeddings, outperforming other AEs and even PCA at low $\beta$ (Fig. 3b). Incorporating prior knowledge fosters disentanglement and improves representation learning. Chromosomal embeddings (Fig. 4b) further show that ontologies are critical for revealing classification-relevant latent dimensions (male–female), even when unrelated to primary variance (tissue).

**Getting the hyperparameterization right for robustness.** A concern with biologically informed networks is robustness to training randomness[30,31], which also applies to ontology-based VAEs. To test this, we calculated latent space correlations across five repetitions with randomized splits, weight initialization, and so on, while varying three hyperparameters: $\beta$, the encoder dropout and the learning rate (Fig. 4c,e).

We expected that (1) $\beta$ limits the latent space flexibility, (2) the dropout mitigates instability from gene-to-pathway overlaps and (3) the learning rate impacts training stability. Results show that robustness across ontology terms is highly nonuniform and dependent. In particular, chromosome-based Ontix shows that larger chromosomes are more robust (see Fig. 4c using TCGA data and compare to Supplementary Fig. 4 using single-cell data). The learning rate had the strongest influence, requiring slower values than typical tuning suggests. This highlights a tradeoff between loss optimization and the reliability of biological insights. In contrast, $\beta$ and the dropout showed no clear robustness patterns (Fig. 4c,e), which is surprising since earlier implementations emphasized high dropout for overlapping gene sets[6,8].

## Data modality translation by X-modalix

Another methodological development using VAEs is cross-modal data translation with AEs that learn separated but aligned latent spaces for each modality, proposed by Yang and Uhler[5,23]. We implemented this cross-modal VAE ('X-modalix') for two modalities, including image data integration, which is unique for this type of AE.

**Efficient translation of expression data to images.** To illustrate capabilities, we studied mainly two datasets and three scenarios. They cover Omics-to-image scenarios where meaningful translation can be visually checked and Omics-to-Omics translation leveraging the relationship of DNA methylation and gene expression.

First, we used TCGA pan-cancer gene expression with handwritten digits (Modified National Institute of Standards and Technology (MNIST), 0–4) assigned to five cancer subtypes to explore loss weighting and hyperparameter configuration (Supplementary Fig. 6). In this synthetic, but verifiable, scenario, we expect the cross-modal VAE to translate the strong gene expression signatures of subtypes to the correct digits. This scenario is an illustration of the capability to translate between molecular–genetic data and image-based descriptions of cells and tissues (microscopy and radiology), which is a major goal bridging genetics and phenotype. It shows that latent space alignment can be reached even for nonpaired multi-omics since we randomly selected image samples for each digit to respective cancer samples and relied on the semisupervised class-based loss term for alignment (Supplementary Fig. 6c). Intentionally, there is no additional link or bias between the modalities, allowing us to prove that relying on class information is sufficient.

Encouraged by this, we trained an X-modalix beyond image translation, examining DNA methylation-to-gene expression on TCGA (Fig. 5a). We observe a well-aligned latent space, and comparison of

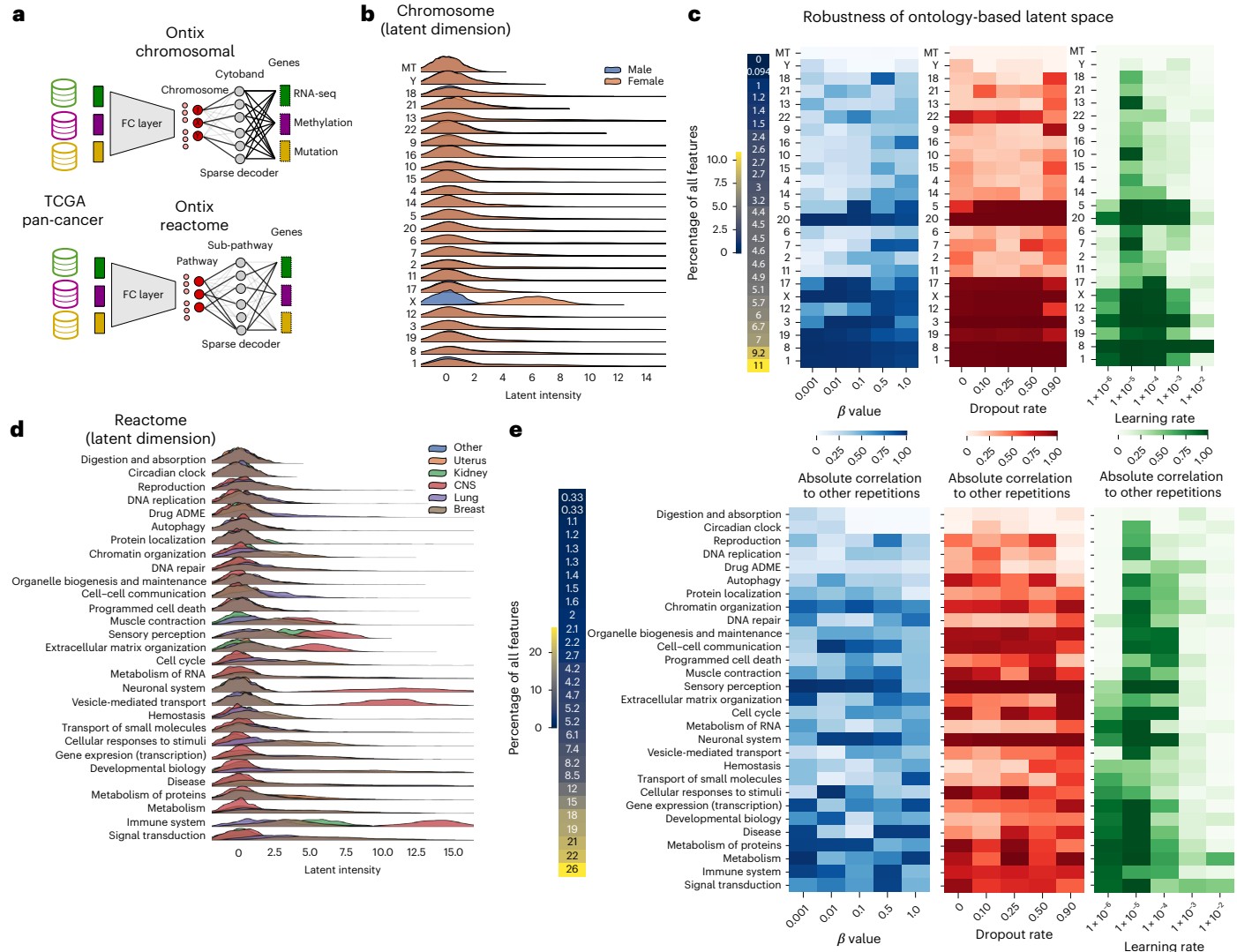

**Fig. 4 | Explainability and robustness of Ontix. a**, Testing of two types of ontologies on the TCGA dataset: chromosomal-based and Reactome-pathway-based latent dimensions. **b,d**, Ridge-line plots of latent intensity distributions, based on selected examples of sample classes (sex and cancer tissue of origin) for chromosomal (**b**) and Reactome-pathway-based (**d**) ontologies. **c,e**, The robustness for chromosomal (**c**) and Reactome-pathway-based (**e**) embeddings in relation to hyperparameters, measured as the mean absolute Pearson correlation between five independent training runs with a randomized data split and weight initialization. FC, full-connected; ADME, absorption, distribution, metabolism and excretion; CNS, central nervous system.

---

reconstructed gene expression from DNA methylation shows high similarity to the original expression in the UMAP representation (Fig. 5b). Together with the similar performance of embeddings on downstream tasks compared to Varix (Fig. 5d), we conclude that biological information is retained in the reconstructed gene expression. However, deeper analysis and sophisticated in silico and in vitro experiments are required to evaluate the translation quality.

Thirdly, we used another real-world scenario from a published dataset of *Caenorhabditis elegans* development where live-cell imaging is available in combination with proteome of hundreds of transcription factors (TFs) (Fig. 5e). By applying our X-modalix to this data, we expect that the TF signature over 260 time points can be translated to images of *C. elegans*.

**Considerations and recommendations to ensure latent space alignment.** For latent space alignment, cross-modal VAEs rely on additional loss terms and training an adversarial network in parallel ('Framework and AE implementation details' section in Methods). Our scenarios cover a synthetic scenario having nonpaired modalities (TCGA RNA sequencing (RNA-seq) and MNIST digits), where alignment relies on an

adversarial loss and semisupervised class-based loss, and paired measurements (TCGA RNA-seq and DNA methylation, *C. elegans*), where alignment relies on an adversarial- and a paired-loss term. Although our framework is not designed to handle unpaired multi-omics data, the first scenario shows that, for X-modalix, we support this by using the class-based loss.

Across scenarios, correct weighting of loss terms proved essential. We recommend prioritizing reconstruction loss, limiting the KL loss to <10% of the reconstruction precision, and distributing the remainder between alignment terms ($\beta$, adversarial loss weight ($\gamma$), paired loss weight ($\delta_{paired}$) and class-based loss weight ($\delta_{class}$)) (Supplementary Fig. 6c and Fig. 5c,g). This setup produced well-aligned latent spaces along classes (cancer subtypes) or time points (developmental stages) (Fig. 5b,f).

This weighting of loss terms leads in the scenarios to well-aligned latent spaces along either classes (cancer subtypes) or time points (developmental stages), as depicted in Fig. 5b,f. Hence, translation of gene expression signatures, or TF proteome data, to images works precisely (Fig. 5). Interestingly, even with only 182 training samples (70%, randomly selected from a total of 260 samples) representing time

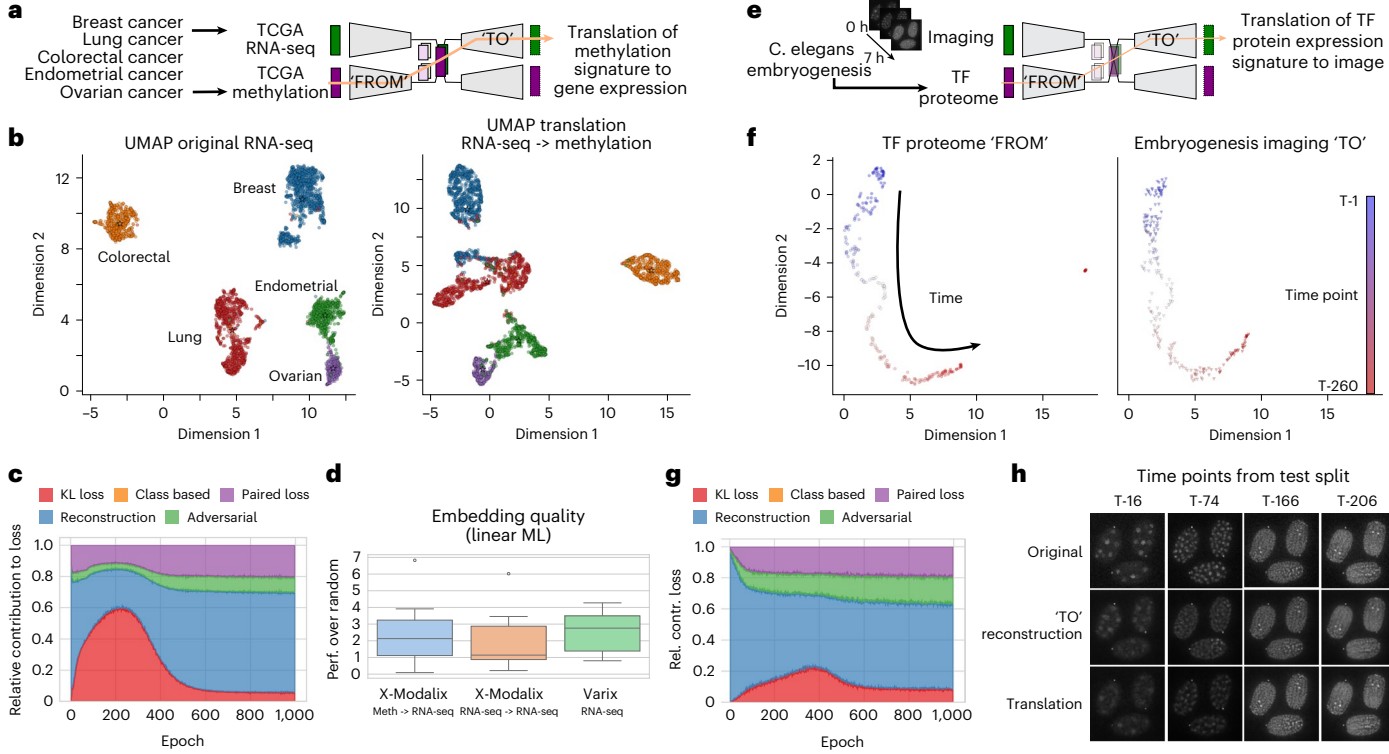

**Fig. 5 | X-modalix scenarios. a**, The translation of DNA methylation to gene expression (RNA-seq) from TCGA. **b**, The 2D-UMAP representations of the original RNA-seq and translated gene expression data from DNA methylation. **c**, The relative contribution of loss during training, calculated on a validation set using the final loss term weightings (not annealed). **d**, The omics-to-omics translation quality, verified by embedding evaluation on downstream linear ML tasks ($n = 12$) calculated as the performance over a random selection of genes as

features (as previously). **e**, The omics-to-image translation scenario from *C. elegans* embryogenesis with 260 time points (labeled as T-1–T-260) spanning over 7 h. **f**, The UMAP representation of the aligned latent spaces for the *C. elegans* scenario. **g**, The relative contribution to loss during training, calculated on a validation set using the final loss term weightings (not annealed). **h**, The image translation capability using proteome (*C. elegans*) embedding of four randomly selected and unseen test samples

points, the reconstruction of images of *C. elegans* from unseen time points is very close to the original data. Further, we observe only a slight reduction in image reconstruction capability by latent space alignment in comparison to a purely image-based VAE (Supplementary Fig. 5).

In comparison, class-based alignment of TCGA RNA-seq with MNIST digits achieved accurate reconstruction of class centers but greater variation across test samples. We attribute this to weaker alignment in peripheral latent space regions, reflecting the absence of biological links between modalities in this synthetic case (Supplementary Fig. 6).

Overall, our results show that cross-modal AEs, including X-modalix, can be trained effectively on paired and even unpaired data (with class labels) to achieve latent space alignment. This highlights their broad potential for multi-omics studies, bridging molecular data layers to gain insights into biological processes and diseases. Beyond biomedicine, X-modalix supports arbitrary images and numerical or categorical data, making it broadly applicable to multimodal research.

## Discussion
In this study, we demonstrated the potential of AUTOENCODIX for benchmarking AEs on multimodal data. We identified important relationships such as the tradeoff between reconstruction capability and latent space coverage, and compared the embedding quality versus PCA and UMAP. PCA's enforced disentanglement improves the embedding quality with more dimensions, while UMAP and vanilla AEs excel in two dimensions. We observe that ontology-based VAEs can improve disentanglement, but require research on architectures, hyperparameters and loss functions[32–34]. Still, PCA remains a strong baseline, with AEs only showing superiority under strong dimension reduction or when

variance unrelated to the task is suppressed. In such cases, stacked VAEs outperform PCA, and ontology-based VAEs excel when embeddings align with the chosen ontology. Prior biological knowledge enables control over disentanglement, reduces model parameters and increases explainability compared to PCA and other methods. Using Optuna[28], we found that tuning only marginally improves reconstruction and embedding quality, highlighting the need for efficient strategies.

Surprisingly, in ontology-based VAEs, tuning sometimes reduces robustness by increasing the sensitivity to training randomness. Since explainability is central to this architecture, robustness is critical for reliable insights. Prior work noted the instability of biologically informed neural networks[30], and dropout was introduced to counter feature–ontology overlaps[6,8]. Our finding that tuning decreases robustness suggests that future strategies must incorporate repeated, randomized training. Benchmarking different ontology schemes will also be essential to maximize both quality and explainability[31]. Importantly, we show that explainability by design can enhance performance, challenging the assumption of a tradeoff between interpretability and accuracy[35]. Beyond this, post hoc feature attribution methods such as SHapley Additive exPlanations[36], Deep Learning Important FeaTures[37], and Local Interpretable Model Agnostic Explanations[38] could extend explainability across AEs.

Cross-modal VAEs for the translation between modalities have immense potential for multi-omics, but also for other domains, in a range of applications such as the imputation of sparse and missing measurements, perturbation studies and recovery of regulatory networks. Training independent but coupled VAEs is especially valuable for unpaired multi-omics, a challenge for most methods. We generalized prior implementations and showed that cross-modal VAEs can

translate reliably between different modalities (images and omics), handle unpaired data with class-loss guidance and perform well even with limited time-series data. Our experiments confirm that carefully weighting loss terms is essential to balance latent space alignment with reconstruction precision. This warrants further research into generalizing hyperparameterization for cross-modal VAEs, as alignment weights cannot be optimized by standard tuning approaches.

In conclusion, our framework provides a standardized, flexible platform for training and evaluating AEs. Its open-source nature enables extensions to new architectures and trends in representation learning. Promising directions include denoising VAEs for time series[19,39,40], single-cell RNA-seq denoising[41], and masked VAEs for imputation[13,42,43], which could advance support for unpaired multi-omics[24]. Core ideas such as explainability by design, classifier co-training and complex loss functions are already implemented. We envision modularizing these components to enable combinations with approaches such as semisupervised and contrastive learning[44].

We commit to long-term support, provide a contribution guide and plan to develop a Python package fully compatible with scVerse. Expanding methods in AUTOENCODIX will foster applications beyond biomedicine and establish the framework as a platform for methodological research and benchmarking.

Finally, we see AUTOENCODIX as a foundation for large-scale and pretrained AE models, the emerging trend in representation learning with single-cell multi-omics[45–48]. Such models could enable fine-tuning with small datasets or even zero-shot applications. This step will greatly broaden the applicability of AEs in biomedical research and beyond.

## Methods

### Framework and AE implementation details
Our open-source framework is implemented in Python and is based on PyTorch for model implementation. It runs on Linux, Windows and MacOS machines, preferably with a graphics processing unit (GPU) offering CUDA (Compute Unified Device Architecture) support, but also on CPU (central processing units) for training of AEs. Finally, we offer via Optuna[28] the possibility for hyperparameter tuning for crucial model and training parameters.

**Preprocessing.** Via our framework, input data are directly preprocessed to merge data modalities in accordance to AE model requirements, and features are filtered and scaled upon user configuration. For the shown experiments and results, if not stated otherwise, we filtered input features (genes) for the most variance, and feature values are scaled by using the in-built scikit-learn function of the standard scaler[49]. We use standard scaling as the default matching mean squared error as reconstruction loss.

Alternatively, we offer the filtering of features also based on the median absolute deviation (MAD, via the SciPy package[50]) and based on correlation using KMedoid clustering to identify representative features among highly correlated feature sets (via the scikit-Learn-extra package[51] with the CLARA algorithm). Further, other scaling methods offered by our framework are directly embedded via scikit-learn, such as MinMax, RobustScaler or MaxAbsScaler.

In addition to filtering and scaling, we support categorical features through one-hot encoding and generate random train, test and validation data splits on the basis of user-specified ratios. For shown experiments, we used a ratio of 60:20:20 if not stated otherwise.

**General AE architectures and implementation.** All AE implementations (Fig. 1) are based on previously published versions and adjusted for standardization as well as generalizability and hyperparameter tuning ability. At the core of all implementations are fully connected layers with the same series of PyTorch standard layers composed of a linear, batch normalization, dropout (default rate 0.1) and ReLU activation layer (if not the final output layer or the last layer before the latent

space layer). The number of layers and hidden layer dimensions in the encoder and decoder are always symmetrical and, in nontuned variants, composed of two layers, each with a hidden layer dimension $h_i$ defined by an encoding factor $e$ (with a nontuned value of 4)

$$h_i = \max \begin{cases} \dfrac{h_{i-1}}{e} \\ L_{\text{dim}} \end{cases}, \tag{1}$$

where $h_{i-1}$ is the layer dimension of the previous layer and $L_{\text{dim}}$ is the user-defined dimension of the latent space.

As in previous studies and implementations of VAEs, the latent space (or the bottleneck) layer is defined by learning in parallel the mean $\mu$ and the logarithmic variance $\log(\sigma^2)$, which follow a Gaussian distribution, which are then combined via the reparameterization trick, such as

$$z = \mu + \epsilon e^{\frac{1}{2}\log(\sigma^2)} \tag{2}$$

where $z$ is the value of the embedding and $\epsilon$ is a randomized value from a normal distribution. The VAE implementation, Varix, is the basis for all other VAE variants (Stackix, Ontix and X-modalix), always using a standard Gaussian as prior.

Training of all AE architectures is based on minimization of the reconstruction loss, with additional loss terms depending on the architecture and described in the following. For the results shown, we employed the mean squared error (MSE) as the reconstruction loss, since it matches our input data characteristics. However, as an alternative, we implemented the option to use the binary cross entropy for the reconstruction loss as well (both via built-in PyTorch functions). For all VAEs, the similarity loss to the normal distribution is determined by the KL divergence as default, although alternatively, users can opt for the maximum mean discrepancy, both of which are weighted via the hyperparameter $\beta$ to control the tradeoff with the reconstruction loss.

All implementations are open source with a special focus on standardization and modularization to enable flexible and easy extension of the framework to other emerging AE architectures and variants.

**Stacked variational AE.** The stacked VAE (Stackix), also referred to as a hierarchical VAE, represents the core idea on which we built our implementation, viz. that separate VAEs are first trained on each of the data modalities before being merged together[22]. The resulting latent space embeddings of each VAE have a dense layer of dimension $d = \frac{1}{8}k_{\text{input},m}$, where $k_{\text{input},m}$ is the number of input features for data modality $m$. These dense layer embeddings are the input for the stacked VAE with final latent dimension $L_{\text{dim}}$ as specified by the user. For training of each VAE of the stacked VAE, the same loss function is used (MSE and KL divergence).

**Ontology-based variational AE.** The ontology-based VAE (Ontix) aims to incorporate biological or other domain knowledge into the architecture. We follow current ideas of knowledge incorporation[6,8] by making the decoder layers sparse and mask according to the feature–ontology term or ontology–ontology term connectivity. In our implementation, as previously, we enforce zero weights in the respective linear layers to mask and restrict the decoder. Current implementations of ontology-based VAEs either are single (linear) layer decoders (VEGA[6] and expiMap[7]) or can have any depth of ontology (OntoVAE[8]). For a tradeoff between simplicity and nonlinearity complexity, our implementation supports either one or two levels of feature–ontology hierarchies. In addition, users can specify additional, fully connected layers if additional dimension reduction (such as two-dimensional (2D) visual representation) is necessary.

**Cross-modal variational AE.** The X-modalix implementation consists of two parts: (1) the method of training two AEs together to enable

the translation functionality and (2) the model architecture itself. For part 1, we closely followed the implementation of Yang et al.[23] and made adjustments for standardization and generalization to match our framework approach. The general idea of a cross-modal VAE is to translate data modality $A$ to data modality $B$ by putting $A$ in the encoder part of the VAE for modality $A$, obtaining the latent representation $z$ and feeding $z$ into the decoder of the VAE for data modality $B$. In our case, we always translated numeric omics data into image data. To ensure a functional translation process, the different AEs need to be trained together to keep the latent spaces aligned.

For step 2, we used the Varix architecture (as described in 'Framework and AE implementation details' section) for numerical data and combined it as described in step 1 with a specialized VAE for image data. The image VAE processes an input image of shape $(C, H, W)$, where $C$ is the number of channels, and $H = W$ (we allow only quadratic images as of now) represents the height and width of the image, respectively, and reconstructs the same-sized output image. The architecture consists of an encoder and a decoder of convolutional layers, connected via a latent space representation.

In line with previous image VAE implementations[23], the encoder consists of five convolutional layers, each with a kernel size of 4, stride of 2 and padding of 1.

We offer pretraining of the image VAE to account for the different complexities of the data modalities, as described by Yang et al.[23], for a user-specified number of epochs before the combined training of both VAEs with latent space alignment. Further, we found that pretraining of the image VAE combined with $\beta$ annealing increases the robustness with respect to posterior collapse during latent space alignment. For latent space alignment, the loss function consists of five different terms: (1) the reconstruction loss and (2) similarity loss, identically for all VAEs, as well as (3) an adversarial loss term, weighted with a parameter $\gamma$, (4) a paired loss term, weighting with $\delta_{paired}$, and (5) a semi-supervised class-based loss term, weighted with $\delta_{class}$. The combination of terms 1–5 is crucial for latent space alignment and subsequent data modality translation.

To obtain the adversarial loss term, a latent space classifier is trained in parallel and learns to classify whether the latent space comes from the image or omics data modality. We used the cross-entropy as a loss function for this term. In the loss term for the VAEs, we then switch the labels of the data modalities in the cross-entropy function. This results in a higher loss when the latent spaces are easy to discriminate for the latent space classifier and thus keeps the latent spaces aligned. The latent space classifier itself is a fully connected neural network consisting of an input layer, two hidden layers and an output layer.

The paired loss term, or the semisupervised class-based loss term, improves the latent space alignment and compactness by minimizing the distance, defined as the mean absolute deviation across all latent dimensions, of a sample across both latent spaces per data modality (paired) or of samples to its class average (center). Since paired measurements of data modalities are rare in multi-omics studies, the class-based approach is an important alternative for real-world applications.

### Datasets

To test and benchmark the capabilities of our framework, we use a number of publicly available datasets to mimic typical use cases and provide a reference for use on other datasets and applications.

The TCGA pan-cancer multi-omics dataset was retrieved as preprocessed files via cBioPortal[52], including patient data for a total of 32 cancer types. We use three data modalities: gene expression data (mRNA-seq V2 RSEM), epigenetic data (methylation HM27 and HM450) and mutational data by combining single-nucleotide variations and copy-number alterations by a simple score, $M_{score,i}$, per gene $i$

$$M_{score,i} = \frac{SNV_i}{CDS_i} + CNA_i \qquad (3)$$

where $SNV_i$ is the number of single-nucleotide variations per gene, $CDS_i$ is the length of the gene (coding region) and $CNA_i$ is the copy number alteration of the gene. In total, the dataset contains 9,267 patient samples with data files across all three modalities.

The sc-Cortex single-cell multi-omic (paired RNA-seq and assay for transposase-accessible chromatin using sequencing (ATAC-seq)) dataset of the developing human cerebral cortex[53] was retrieved as preprocessed h5ad files from CZ CELLxGENE Discover[54], including a total of 45,549 cells (nuclei).

*C. elegans* embryogenesis microscopic images and proteomic data, based on a transcription factor reporter atlas, were used for the cross-modal VAE[55]. In addition, MNIST handwritten digit images retrieved via the Keras package[56] were used in combination with the TCGA gene expression data.

A summary of the datasets and key figures is given in Supplementary Table 1.

### Experiments

All of the scripts for the shown experiments and the visualizations of their results can be accessed and reproduced via our GitHub repository (https://github.com/jan-forest/autoencodix-reproducibility). In particular, the benefit of our pipeline is the configuration via YAML files, providing reproducibility by design as well as enabling usage on new datasets.

**Impact of $\beta$ annealing and training of variational AEs.** As a baseline representation, five Varix with two-dimensional latent space, which enables direct visualization, were trained on all three data modalities of the TCGA dataset with 2,000 features (genes) of each modality. During training over 1,000 epochs, the weighting factor $\beta$ of the KL divergence loss term was annealed with the default function using simple logistic annealing

$$\beta(epoch) = \frac{\beta_{final}}{1 + e^{-B(epoch - M \, total \, epochs)}} \qquad (4)$$

which provides robustness of VAE training with regard to posterior collapse, as shown previously[57,58]. For comparison, five Varix where trained with different $\beta_{final}$ values of 0, 0.01, 0.1, 1 and 10.

When using VAEs as generative models, good latent space coverage would be critical to generate synthetic and meaningful samples in a broad range. To define the latent space coverage, we assume the following, given that the samples are perfectly distributed among latent space dimensions: If the latent space is only one-dimensional, one could have a grid with $b$ bins where $b$ equals the number of samples ($n$) and when perfectly distributed each bin would contain exactly one sample. If the latent space is two-dimensional, each latent dimension can be binned with $b = \sqrt[2]{n}$ or, in general, $b = \sqrt[l_{dim}]{n}$ for any latent dimension. Bins to check coverage are then defined as a grid between minimal and maximal values for each latent dimension across samples. The coverage of a latent dimension $l_i$ is then defined as the fraction of bins containing at least one sample among the total number of bins $b$. In Fig. 2b, the average coverage over the two dimensions is shown. For Gaussian distributions, a maximal coverage of 0.5 can be expected.

**Comparison of AE architectures and the impact of tuning for multi-omics data integration.** For benchmarking of the available AE architectures of our framework, we used the TCGA and sc-Cortex dataset and trained AEs on all combinations of up to three data modalities with a uniform configuration and parameters, as outlined in Supplementary Table 2. Furthermore, key hyperparameters (Supplementary Table 2) have been tuned via Optuna as a built-in option of our framework for the same number of epochs and 50 optimization trials. Datasets were split into training, validation and test sets in the ratio of 60:20:20.

Since the reconstruction loss, calculated as the MSE loss, is not invariant to input size and comparable across datasets and configurations, we report the reconstruction capability as the explained variance $R^2$ between the input and reconstruction per sample. Final reported values are the mean $R^2$ over all test samples and are directly related to the MSE loss.

The ontology-based VAE (Ontix) was constructed and trained using biological pathways from the Reactome database[29]. The highest level of pathway hierarchy (top level) was used to define the 29 dimensions in the latent space. As for the hidden layer, we used the third highest level in the pathway hierarchy to define the first ontology level between the genes and each pathway. When compared to other AE architectures with $L_{dim} = 2$ or 8, a third hidden layer (fully connected) is introduced for dimensionality reduction and comparability to other architectures with $L_{dim} < 29$.

To evaluate embedding (AEs, PCA or UMAP) performance, available biological and clinical annotations of samples were used to train regression or classification tasks using three different ML algorithm classes (linear or logistic regression, support vector machines with radial kernel, and random forests) using scikit-learn implementations. The performance metrics used were the area under the receiver operating curve for classification tasks and the explained variance $R^2$ for regression tasks. To enable comparability across tasks with varying difficulty, the performance using randomly selected features as the simplest mean of dimension reduction (five repetitions) was determined.

The performance of embeddings over randomly selected features ($z_{i,m}$) was then calculated and determined based on Z-score normalization to account for the mean (avg) and standard deviation (s.d.) across five ($r \in [1, 5]$) randomly selected features $\eta_{r,i,m}$ for the $i$th classification or regression task and $m$th ML algorithm

$$z_{i,m} = \frac{\theta_{i,m} - \text{avg}(\eta_{r,i,m})}{\text{s.d.}(\eta_{r,i,m})}, \tag{5}$$

where $\theta_{i,m}$ is the performance of the embedding (AE, PCA or UMAP) as either the explained variance ($R^2$, for regression tasks) or the area under the receiver operating curve (for classification tasks). We apply this Z-score normalization to take into account that some ML tasks, such as cell type classification, are easier and expected improvements from embeddings are much smaller than for hard prognostic tasks, such as months of survival.

**Analysis and robustness of ontology-based AEs.** To test the reliability and robustness of trained latent space intensities associated with ontologies, we conducted five repetitions per hyperparameter and ontology combination. As ontologies, we used Reactome pathways (as described in previous sections) and chromosomal positions where latent space dimensions represent human chromosomes (including X, Y and mitochondrial genes). As a hidden layer, we used information via Ensembl BioMart[59] about cytobands to infer substructures of human chromosomes (version GRCh38.p14) where genes (features) are located.

The baseline hyperparameter configuration is $\beta = 0.1$, $p_{dropout} = 0.5$ and a learning rate of $1 \times 10^{-4}$. To gain insights into the impact of these parameters on robustness, we varied each parameter individually for five different values and performed five training repetitions. Each training is randomized across the whole pipeline including train, test and validation splits or model weight initialization.

As a measurement of robustness, we calculate the average Pearson correlation across all repetitions for each latent space dimension representing an ontology. Since we are not interested in qualitative robustness and have not reversed orders in latent spaces, we calculate the absolute value of correlation before averaging over repetition combinations.

**Scenarios of cross-modal AE.** To showcase the capabilities of X-modalix, we ran three experiments: First, we investigated the cross-modal translation from expression data of transcription data to embryonic images of the *C. elegans* organism. Second, we synthetically coupled TCGA omics data with handwritten digits (MNIST) to learn more about hyperparameter configuration. Third, we studied the omics-to-omics scenario, leveraging TCGA data from DNA methylation and gene expression (RNA-seq).

The first scenario uses a protein expression atlas of TFs at single-cell resolution, collected by Ma et al.[55], mapped onto developmental cell lineages during *C. elegans* embryogenesis as well as corresponding four-dimensional (4D) imaging of embryogenesis. The authors took images every 75 s during embryonic development. At each time point, three slide positions were scanned across 30 focal ($z$) planes with a spacing of 1 μm between planes. This results in 300 images (one for each time point) per TF. Each image has 30 z channels and two color channels, which we processed as follows:

We used, for simplicity, only one channel (red) and applied maximum intensity projection across the z-dimension, resulting in 512 × 512 images per time point and TF. Pixel values are normalized to a range of 0–255. To reveal the spatiotemporal development of *C. elegans*, one TF with high and homogeneous expression is sufficient instead of aggregating across all TFs. Hence, TF gene *ALY-2* was selected as a representative TF on the basis of its ubiquitous signal across all time points and cell states (individual expression patterns are accessible via the web service of the authors (https://dulab.genetics.ac.cn/TF-atlas/Gene.html)).

TF proteomic expression was aggregated and normalized across all strains and time points. We normalized the expression value of each TF by the sum of the expression of all TFs at the same time point to retrieve a proteomic signature rather than the absolute expression across time points. After removing missing values (dropping TFs with more than 90% missing values and then removing all time points with missing values), the dataset had 260 time points (samples) and 290 TFs (features).

We used a 70% train, 10% validation and 20% test split for the dataset with random selection across time points. The image VAE was pretrained for 100 epochs, including logistic $\beta$-annealing. A complete list of hyperparameters is presented in Supplementary Table 2. We optimized the loss term weights $\beta$, $\gamma$, $\delta_{paired}$ and $\delta_{class}$ manually to achieve a balance between reconstruction capability and latent space alignment.

For the second cross-modal experiment, we mapped MNIST handwritten digit images to the five most common TCGA cancer types. We processed the TCGA dataset as before, using the gene expression data modality only. We obtained the MNIST dataset from the Keras package[56] and mapped randomly selected image samples of digits '0' to '4' to the most frequent cancer types (breast cancer for 0, non-small cell lung cancer for 1, colorectal cancer for 2, endometrial cancer for 3 and ovarian epithelial tumor for 4). The resulting dataset consisted of 3,529 images, with 1,065 images for the digit '0', 991 images for '1', 589 for '2', 584 for '3' and 300 for the digit '4'. We evaluated the translation as in the *C. elegans* example. Additionally, we defined centers of cancer subtype classes by calculating the latent space embeddings of all samples of each class for either the RNA-seq VAE encoder ('FROM') or the image-based VAE ('TO'). When retrieving center representations from the image-based VAE decoder ('TO'), for each class and each latent dimension, the median embedding value across samples was taken as the input for the decoder. A quantitative evaluation of the image reconstruction quality is given in Supplementary Fig. 5 for both scenarios.

In a third experiment, we evaluated the utility of the X-modalix model for multimodal omics data. We had two objectives with this experiment: (1) to evaluate the translation capability for the nonimage case and (2) to evaluate whether the alignment of latent spaces brings advantages in downstream analysis. For these goals, we trained the model to translate from DNA methylation data to gene expression data. To ensure consistency, we used the same gene expression TCGA data as for the MNIST experiment and included sample-matched DNA methylation data. Additionally, we trained a Varix model with gene

expression data as a single data modality for reference. We then evaluated the reconstruction quality and biological information retention by comparing the UMAP representation of the original RNA-seq data with the translated gene expression data from methylation. Analogously to previous analysis, we calculated the embedding quality via selected ML tasks with linear models and calculated the performance over random feature selection.

Supplementary Table 2 summarizes the hyperparameters and configuration for all these experiments.

### Reporting summary
Further information on research design is available in the Nature Portfolio Reporting Summary linked to this article.

## Data availability
No data were generated for this research. Publicly available datasets were used for framework development and analysis: the TGCA pan-cancer multi-omics dataset[60], retrieved as preprocessed files via cBioPortal[52]; the sc-Cortex single-cell multi-omic (paired RNA-Seq and ATAC-Seq) dataset of the developing human cerebral cortex[53], retrieved as preprocessed h5ad files from CZ CELLxGENE Discover[54]; *C. elegans* embryogenesis microscopic images and proteomic data, based on a transcription factor reporter atlas from ref. 55. The collected and preformatted data, as described in 'Datasets' section in Methods, is available via Zenodo https://doi.org/10.5281/zenodo.15518831(ref. 61). Source data are provided with this paper.

## Code availability
A reproducibility code repository with all relevant scripts is available via GitHub at https://github.com/jan-forest/autoencodix-reproducibility and via Zenodo at https://doi.org/10.5281/zenodo.17190167 (ref. 62). Our software tool is available via GitHub at https://github.com/jan-forest/autoencodix.

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

## Acknowledgements

M.J.J. acknowledges support by the German Federal Ministry of Education and Research under funding code 03ZU1111MB (SaxoCellOmics) as part of the Clusters4Future cluster SaxoCell. J.E., N.S. and N.J. acknowledge financial support by the Federal Ministry of Education and Research of Germany and by Sächsische Staatsministerium für Wissenschaft, Kultur und Tourismus in the program Center of Excellence for AI-research Center for Scalable Data Analytics and Artificial Intelligence Dresden/Leipzig with project identification no. ScaDS.AI. D.P. acknowledges support by the Sächsische Staatsministerium für Wissenschaft, Kultur und Tourismus under the framework of ERA PerMed (GRAMMY, 2019-275 and MIRACLE, 2021-055). N.S. is supported by the BMBF (Federal Ministry of Education and Research) through ACONITE (01IS22065) and by the European Union and the Free State of Saxony through BIOWIN. Further, we thank I. Kreller and all colleagues from ScaDS.AI Dresden/Leipzig for framework testing and critical feedback. N.J. recognizes the editing guidance from the Kather group, EKFZ, TU Dresden. The TCGA dataset used in our experiments was generated by the TCGA Research Network (https://www.cancer.gov/tcga).

## Author contributions

M.J.J. and J.E. mainly developed the software, conceived the study design and data analysis. N.J. and D.P. supported the development of the software and data analysis. N.S. supported the study design and framework conceptualization. M.J.J. and J.E. wrote the paper draft, and all authors contributed to writing the final manuscript, including visualizations and repository preparation.

## Funding

## Competing interests

The authors declare no competing interests.

## Additional information

**Correspondence and requests for materials** should be addressed to Maximilian Josef Joas or Jan Ewald.

# Reporting Summary

## Statistics

For all statistical analyses, confirm that the following items are present in the figure legend, table legend, main text, or Methods section.

| n/a | Confirmed | |
|---|---|---|
| ☐ | ☒ | The exact sample size (*n*) for each experimental group/condition, given as a discrete number and unit of measurement |
| ☐ | ☒ | A statement on whether measurements were taken from distinct samples or whether the same sample was measured repeatedly |
| ☐ | ☒ | The statistical test(s) used AND whether they are one- or two-sided *Only common tests should be described solely by name; describe more complex techniques in the Methods section.* |
| ☐ | ☒ | A description of all covariates tested |
| ☐ | ☒ | A description of any assumptions or corrections, such as tests of normality and adjustment for multiple comparisons |
| ☐ | ☒ | A full description of the statistical parameters including central tendency (e.g. means) or other basic estimates (e.g. regression coefficient) AND variation (e.g. standard deviation) or associated estimates of uncertainty (e.g. confidence intervals) |
| ☐ | ☒ | For null hypothesis testing, the test statistic (e.g. *F*, *t*, *r*) with confidence intervals, effect sizes, degrees of freedom and *P* value noted *Give P values as exact values whenever suitable.* |
| ☐ | ☒ | For Bayesian analysis, information on the choice of priors and Markov chain Monte Carlo settings |
| ☐ | ☒ | For hierarchical and complex designs, identification of the appropriate level for tests and full reporting of outcomes |
| ☐ | ☒ | Estimates of effect sizes (e.g. Cohen's *d*, Pearson's *r*), indicating how they were calculated |

*Our web collection on statistics for biologists contains articles on many of the points above.*

## Software and code

Policy information about availability of computer code

| Data collection | We provide a reproducibility code repository with all relevant scripts under: https://github.com/jan-forest/autoencodix-reproducibility The collected data from sources which are specified in the main text is stored via Zenodo under: https://zenodo.org/records/13691753 |
|---|---|
| Data analysis | All computational experiments and analysis including plot generation can be reproduced using the code under our reproducibility repository: https://github.com/jan-forest/autoencodix-reproducibility |

For manuscripts utilizing custom algorithms or software that are central to the research but not yet described in published literature, software must be made available to editors and reviewers. We strongly encourage code deposition in a community repository (e.g. GitHub). See the Nature Portfolio guidelines for submitting code & software for further information.

## Data

Policy information about availability of data

All manuscripts must include a data availability statement. This statement should provide the following information, where applicable:
- Accession codes, unique identifiers, or web links for publicly available datasets
- A description of any restrictions on data availability
- For clinical datasets or third party data, please ensure that the statement adheres to our policy

No data have been generated for this research. Publicly available datasets have been used for framework development and analysis. The collected and pre-formatted data,

as described under the section Datasets, is stored and hosted via Zenodo https://zenodo.org/records/15518831. Source data are provided with this paper.

## Human research participants

Policy information about studies involving human research participants and Sex and Gender in Research.

| | |
|---|---|
| Reporting on sex and gender | Does not apply. Only public available data sets have been used and no data have been generated via own studies. |
| Population characteristics | Does not apply. Only public available data sets have been used and no data have been generated via own studies. |
| Recruitment | Does not apply. Only public available data sets have been used and no data have been generated via own studies. |
| Ethics oversight | Does not apply. Only public available data sets have been used and no data have been generated via own studies. |

Note that full information on the approval of the study protocol must also be provided in the manuscript.

## Field-specific reporting

Please select the one below that is the best fit for your research. If you are not sure, read the appropriate sections before making your selection.

☒ Life sciences    ☐ Behavioural & social sciences    ☐ Ecological, evolutionary & environmental sciences

For a reference copy of the document with all sections, see nature.com/documents/nr-reporting-summary-flat.pdf

## Life sciences study design

All studies must disclose on these points even when the disclosure is negative.

| | |
|---|---|
| Sample size | Does not apply. Only public available data sets have been used and no data have been generated via own studies. |
| Data exclusions | Does not apply. Only public available data sets have been used and no data have been generated via own studies. |
| Replication | Does not apply. Only public available data sets have been used and no data have been generated via own studies. |
| Randomization | Does not apply. Only public available data sets have been used and no data have been generated via own studies. |
| Blinding | Does not apply. Only public available data sets have been used and no data have been generated via own studies. |

## Reporting for specific materials, systems and methods

We require information from authors about some types of materials, experimental systems and methods used in many studies. Here, indicate whether each material, system or method listed is relevant to your study. If you are not sure if a list item applies to your research, read the appropriate section before selecting a response.

### Materials & experimental systems

| n/a | Involved in the study |
|---|---|
| ☒ | ☐ Antibodies |
| ☒ | ☐ Eukaryotic cell lines |
| ☒ | ☐ Palaeontology and archaeology |
| ☒ | ☐ Animals and other organisms |
| ☒ | ☐ Clinical data |
| ☒ | ☐ Dual use research of concern |

### Methods

| n/a | Involved in the study |
|---|---|
| ☒ | ☐ ChIP-seq |
| ☒ | ☐ Flow cytometry |
| ☒ | ☐ MRI-based neuroimaging |

