## [Peer review file · Nature Computational Science]

AUTOENCODIX: A generalized and versatile framework to train and evaluate autoencoders for biological representation learning and beyond

Corresponding Author: Dr Jan Ewald

This manuscript has been previously reviewed at another journal that is not operating a transparent peer review scheme. The manuscript was considered suitable for publication without further review at Nature Computational Science.

Version 0:

Decision Letter:

** Please ensure you delete the link to your author homepage in this e-mail if you wish to forward it to your co-authors. **

Dear Dr Ewald,

Your manuscript "A generalized and versatile framework to train and evaluate autoencoders for biological representation learning and beyond: AUTOENCODIX" has now been seen by 2 referees, whose comments are appended below. You will see that while they find your work of interest, they have raised points that need to be addressed before we can make a decision on publication.

The referees' reports seem to be quite clear. Naturally, we will need you to address *all* of the points raised.

While we ask you to address all of the points raised, the following points need to be substantially worked on:

- A discussion of the role of paired versus unpaired data is missing. Please discuss how this framework would manage to handle a variety of pairing settings.
- Please provide simpler tutorials that can be easily run on a laptop.
- Please position the work in comparison to contrastive learning approaches (Richter, Till, et al. "Delineating the effective use of self-supervised learning in single-cell genomics". Nature Machine Intelligence (2024): 1-11.) that create an interesting latent space.
- In Fig. 2B, the β parameter is varied every 400 epochs to explore disentanglement versus reconstruction trade-offs. However, this experimental design conflates the effects of increased training time and changing β . Since reconstruction typically improves with training time while increasing β penalizes reconstruction through KL divergence, it becomes difficult to isolate the effect of β alone. Please address this.
- As shown in Fig. 2C, Ontix consistently underperforms compared to Varix in predictive tasks. Please clarify when and why such trade-offs are acceptable
- Please emphasize case studies or visualizations where Ontix reveals biologically meaningful patterns.
- Given that TCGA contains real multi-modal data, it is unclear why a real modality-to-modality translation was not used to evaluate the model's utility.

Please use the following link to submit your revised manuscript and a point-by-point response to the referees' comments (which should be in a separate document to any cover letter):

Link Redacted

** This url links to your confidential homepage and associated information about manuscripts you may have submitted or be reviewing for us. If you wish to forward this e-mail to co-authors, please delete this link to your homepage first. **

To aid in the review process, we would appreciate it if you could also provide a copy of your manuscript files that indicates your revisions by making use of Track Changes or similar mark-up tools. Please also ensure that all correspondence is

marked with your Nature Computational Science reference number in the subject line.

In addition, please make sure to upload a Word Document or LaTeX version of your text, to assist us in the editorial stage.

To improve transparency in authorship, we request that all authors identified as 'corresponding author' on published papers create and link their Open Researcher and Contributor Identifier (ORCID) with their account on the Manuscript Tracking System (MTS), prior to acceptance. ORCID helps the scientific community achieve unambiguous attribution of all scholarly contributions. You can create and link your ORCID from the home page of the MTS by clicking on 'Modify my Springer Nature account'. For more information please visit www.springernature.com/orcid.

We hope to receive your revised paper within three weeks. If you cannot send it within this time, please let us know.

Best regards,

Ananya Rastogi, PhD
Senior Editor
Nature Computational Science

Reviewers comments:

Reviewer #1 (Remarks to the Author):

The main contribution of this paper is to bring very different approaches using auto-encoders under the same framework and to compare their respective performance in different contexts and under different sets of parameter values. Beyond their ability to produce nonlinear low-dimensional representations of complex datasets, auto-encoders are useful for their generative power and also for their ability to incorporate prior knowledge for interpretability, making them versatile tools with high applicability in multi-omics data analysis. The paper is well written with clear objectives.

The paper provides a very useful broad benchmark between auto-encoder architectures for common tasks such as embedding performance for prognostic clinical models, data modality translation, and explainability by design with ontology-based AEs.

The conclusions that reconstruction capability and embedding quality are not directly related are particularly interesting, and the prescriptions for parameter values are quite useful in practice.

The architectures presented are representative of the state of the art in the field.

Main comments:

1) However, I am missing a discussion of the role of paired versus unpaired data. Several recent papers have discussed variations such as vertical, diagonal, or mosaic integration (Argelaguet, Ricard, et al. "Computational principles and challenges in single-cell data integration." *Nature Biotechnology* 39.10 (2021): 1202-1215, He, Zhen, et al. "Mosaic integration and knowledge transfer of single-cell multimodal data with MIDAS." *Nature Biotechnology* (2024): 1-12, Samaran, Jules, Gabriel Peyré, and Laura Cantini. "scConfluence: single-cell diagonal integration with regularized inverse optimal transport on weakly connected features." *Nature Communications* 15.1 (2024): 7762, Cao, Yichuan, et al. "scButterfly: a versatile single-cell cross-modality translation method via dual-aligned variational autoencoders." *Nature Communications* 15.1 (2024): 2973. Tang, Xin, et al. "Explainable multi-task learning for multi-modality biological data analysis." *Nature Communications* 14.1 (2023): 2546.). How would this framework manage to handle this variety of pairing settings?

2) The code is still quite expert, and the question remains how this kind of approach will be widely adopted by bioinformaticians compared to implementing their own directly. Could you provide simpler tutorials that can be easily run on a laptop? Also, would it be possible to make this framework compatible with the scVerse suite? This would help the adoption rate, as scVerse has recently become the standard in bioinformatics.

3) Since, as a result of your parameter analysis, reconstruction capability is not directly correlated with performance on downstream tasks in the latent space, one may wonder whether AE is necessarily the best architecture. Can you position your work in comparison to contrastive learning approaches (Richter, Till, et al. "Delineating the effective use of self-supervised learning in single-cell genomics". *Nature Machine Intelligence* (2024): 1-11.) that create an interesting latent space? Would your framework be easy to extend to this setting?

Minor comments:

Line 111: please provide more details on the latent space density and correlation of the latent dimension. For example, how are these measures normalized? What values would help characterize the quality of an embedding?

Line 242: Please provide more details on how the training set of images/transcriptomic profile is generated. Are the predicted images time points that are part of the training dataset?

Line 464: How did you decide to focus on only one gene? And why this one? Does it have a monotonically changing dynamic over the movie?

Reviewer #1 (Remarks on code availability):

The code is hosted on a well documented GitHub with a clear README file and several tutorials as python notebooks designed to help first-time users get familiar with the library.

I have noted however several difficulty when working with MACOS

1) I couldn't find the virtual environment venv-gallia

2) several commands are for linux only like `lvmget`

Please make sure that the tutorial notebooks can be run smoothly from any operating system, not only linux.

Reviewer #2 (Remarks to the Author):

This paper introduces AUTOENCODIX, a unified framework for training, evaluating, and comparing a broad range of autoencoder-based models tailored for biological data. The authors benchmark several models, including Vanillix (standard AE), Varix (β -VAE), Stackix (hierarchical VAE), Ontix (ontology-based VAE), and X-Modalix (cross-modal VAE). The inclusion of Ontix and X-Modalix is particularly innovative and promising for enabling biologically interpretable and modality-bridging representations.

The models are evaluated on both single-cell RNA-seq data and TCGA multi-omics datasets. The authors assess model performance based on two main criteria: (1) reconstruction accuracy and (2) performance on downstream regression and classification tasks using linear models, random forests, and SVMs. Additionally, the authors compare AE-derived embeddings with PCA and UMAP, and use Z-score normalized performance over random feature baselines to enable fair, dimension-aware comparison.

The main concerns are surrounding the process of comparisons:

On comparison to β VAE. In Fig. 2B, the authors vary the β parameter every 400 epochs to explore disentanglement versus reconstruction trade-offs. However, this experimental design conflates the effects of increased training time and changing β . Since reconstruction typically improves with training time while increasing β penalizes reconstruction through KL divergence, it becomes difficult to isolate the effect of β alone. This should be addressed by either:

- Including fixed- β baselines trained over the same number of epochs
- Visualizing loss trajectories explicitly separated by β rather than epoch count

On comparison to Stackox (hierarchical AE). The concept of hierarchical stacked autoencoders is well-motivated, but the paper lacks clarity regarding:

- The loss function(s) used for the stacked architecture
 - How the model handles different data types, e.g., categorical genotypes vs. continuous expression data
 - Whether all modalities share the same latent space prior, and how concatenated latent representations are regularized
- On comparison to Ontix (Ontology-informed AE): Ontix is a conceptually strong model that incorporates domain knowledge (e.g., pathways, chromosomes) into the architecture to provide interpretable latent features. However:

- As shown in Fig. 2C, Ontix consistently underperforms compared to Varix in predictive tasks. The authors should clarify when and why such trade-offs are acceptable (e.g., clinical, regulatory, or exploratory discovery).
- It remains unclear:

- o Whether the number of latent dimensions corresponds to the number of ontology terms
- o How overlapping gene-pathway memberships are handled
- o How many ontology layers are used, and whether one-to-one mappings exist between latent dimensions and biological categories

Despite the lower predictive performance, Ontix's interpretability potential is important. The authors should emphasize case studies or visualizations where Ontix reveals biologically meaningful patterns (e.g., sex-related differences on chromosome X, as briefly mentioned).

On the comparison to X-Modalix (Cross-modal AE): The motivation behind X-Modalix — to enable translation across omics modalities — is compelling. However, the main demonstration (gene expression \rightarrow MNIST digits) is synthetic and biologically unmotivated.

Given that TCGA contains real multi-modal data (expression, methylation, miRNA, etc.), it is unclear why a real modality-to-modality translation (e.g., expression \leftrightarrow methylation) was not used to evaluate the model's utility. At minimum, the authors should justify this choice and address whether technical limitations (e.g., data sparsity, alignment) prevented a more biologically relevant experiment.

Here are additional concerns:

1. Missing R^2 for AE Reconstruction:

While the authors use R^2 for CNN regression (Table S4), reconstruction quality for autoencoders is reported only via raw loss, which is scale- and sample-size dependent. R^2 is a scale-invariant and interpretable measure of variance explained. Reporting R^2 for each AE variant would provide much-needed clarity in comparing reconstruction effectiveness.

2. Strong PCA Baseline (Fig. 2D):

PCA with 29 components performs comparably — or better — than many AE models. Given its simplicity and interpretability,

the authors should explicitly discuss when PCA is insufficient and whether AE models truly offer added value for downstream biological tasks.

3. Latent Dimension vs. Classifier Interaction (Fig. S2):

The paper reports that larger latent spaces improve performance for linear models, but degrade performance for SVMs and RFs. This trend deserves deeper analysis:

- Is this due to noise accumulation, feature redundancy, or curse of dimensionality?
- Would feature selection or sparse priors mitigate this?

Understanding this interaction is critical for practical deployment of latent embeddings.

Summary:

AUTOENCODIX is a potentially valuable contribution that brings a unified, modular framework to biological representation learning. It covers a wide landscape of model designs and offers a useful experimental protocol. However, several evaluation design issues, including incomplete metric reporting, overreliance on synthetic data, and insufficient biological justification for some models, weaken the paper's clarity and impact.

With clarification, more complete biological evaluation, and consistent R^2 -based metrics, this framework might serve as a benchmark platform for future bio-AE research.

Reviewer #2 (Remarks on code availability):

Code is well maintained.

Version 1:

Decision Letter:

**** Please ensure you delete the link to your author homepage in this e-mail if you wish to forward it to your co-authors. ****

Dear Dr Ewald,

Your manuscript "A generalized and versatile framework to train and evaluate autoencoders for biological representation learning and beyond: AUTOENCODIX" has now been seen by 2 referees, whose comments are appended below. You will see that while they find your work of interest, they have raised points that need to be addressed before we can make a decision on publication.

The referees' reports seem to be quite clear. Naturally, we will need you to address **all** of the points raised.

While we ask you to address all of the points raised, the following points need to be substantially worked on:

- In the result section corresponding to cross-modal VAE X-modalix, the model is evaluated on unpaired data (only at the class level). It is unclear how the artificial pairing is created, is it a random shuffle of samples of similar classes that are paired randomly? Please discuss this.

Please use the following link to submit your revised manuscript and a point-by-point response to the referees' comments (which should be in a separate document to any cover letter):

Link Redacted

**** This url links to your confidential homepage and associated information about manuscripts you may have submitted or be reviewing for us. If you wish to forward this e-mail to co-authors, please delete this link to your homepage first. ****

To aid in the review process, we would appreciate it if you could also provide a copy of your manuscript files that indicates your revisions by making use of Track Changes or similar mark-up tools. Please also ensure that all correspondence is marked with your Nature Computational Science reference number in the subject line.

In addition, please make sure to upload a Word Document or LaTeX version of your text, to assist us in the editorial stage.

To improve transparency in authorship, we request that all authors identified as 'corresponding author' on published papers create and link their Open Researcher and Contributor Identifier (ORCID) with their account on the Manuscript Tracking System (MTS), prior to acceptance. ORCID helps the scientific community achieve unambiguous attribution of all scholarly contributions. You can create and link your ORCID from the home page of the MTS by clicking on 'Modify my Springer Nature account'. For more information please visit www.springernature.com/orcid.

We hope to receive your revised paper within three weeks. If you cannot send it within this time, please let us know.

Best regards,

Ananya Rastogi, PhD
Senior Editor
Nature Computational Science

Reviewers comments:

Reviewer #1 (Remarks to the Author):

The authors have answered most of my comments. They mention a bug in the implementation that has been identified between the previous and current version of the article, however the main messages are not affected.

I still have an important remark

The authors have clarified that their architectures are meant to be used on paired data. In particular they write in the introduction "Currently, our implemented architectures rely on paired measurements which are often difficult to obtain in multi-omics settings". However, in the result section corresponding to cross-modal VAE X-modalix, the model is evaluated on unpaired data (only at the class level). They write "Our scenarios cover two cases either having non-paired data modalities .. and paired measurements. "Notably, although our framework is not designed to handle unpaired multi-omics data, the first scenario shows that for X-modalix we can support this by using the class-based loss." It is unclear how the artificial pairing is created, is it a random shuffle of samples of similar classes that are paired randomly? In that case, how is the shuffling controlled and is it possible that it would bias the data? I am wondering about the relevance of the TCGA to MNIST experiments. The relationship between the two modalities is highly artificial as there is in principle no mechanism linking RNASeq to written digits. This type of example for illustrating the translation ability of cross-modal autoencoders might even induce suspicion on the biological relevance of this task. Previous authors have warned that non-linear methods such as UMAP can hinder biological relevance, e.g. "The specious art of single-cell genomics" <https://journals.plos.org/ploscompbiol/article?id=10.1371/journal.pcbi.1011288> . I am slightly worried that the example, RNASeq to MNIST translation proposed by the authors might lead to similar worry. Since the TCGA RNASeq to methylation use case works well (and is performed on paired data), I am wondering if it wouldn't be more relevant to include in the main text instead of the RNASeq to MNIST use case.

Similarly, I do not understand the statement that "We intentionally focus on Omics-to-image scenarios because meaningful translation can be visually checked whereas Omics-to-Omics translation typically needs experimental validation to prove biological correctness". I do not agree that looking at realistic images is sufficient to assess validity of the approach, especially when the link between the RNASeq to image is largely artificial, and if anything, shows the ability of the method of performing clustering. Label being associated to the average image of the class. In the Yang et al. paper that the authors use as a starting point, the assumption that the various modalities are governed by the same underlying latent variable is clearly stated and should be a prerequisite for this type of approaches.

Reviewer #1 (Remarks on code availability):

The code is well maintained. I tried to run the quick start tutorial https://github.com/jan-forest/autoencodix/blob/main/Tutorials/Quick_Start.ipynb - however several limitations appeared with Makefile - due probably to the way indentation are encoded in the text editor used by the authors. Please double check on macOS.

Reviewer #2 (Remarks to the Author):

All the comments has been addressed, I have no further comments.

Reviewer #2 (Remarks on code availability):

The code are good to use.

Version 2:

Decision Letter:

Our ref: NATCOMPUTSCI-25-0037B

13th August 2025

Dear Dr. Ewald,

Thank you for submitting your revised manuscript "A generalized and versatile framework to train and evaluate autoencoders for biological representation learning and beyond: AUTOENCODIX" (NATCOMPUTSCI-25-0037B). It has now been seen by the original referees and their comments are below. The reviewers find that the paper has improved in revision, and therefore we'll be happy in principle to publish it in Nature Computational Science, pending minor revisions to satisfy the referees' final requests and to comply with our editorial and formatting guidelines.

TRANSPARENT PEER REVIEW

Nature Computational Science offers a transparent peer review option for original research manuscripts. We encourage increased transparency in peer review by publishing the reviewer comments, author rebuttal letters and editorial decision letters if the authors agree. Such peer review material is made available as a supplementary peer review file. **Please remember to choose, using the manuscript system, whether or not you want to participate in transparent peer review.**

Thank you again for your interest in Nature Computational Science. Please do not hesitate to contact me if you have any questions.

Sincerely,

Kaitlin McCardle, PhD
Senior Editor
Nature Computational Science

ORCID

Reviewer #1 (Remarks to the Author):

Thank you, the authors have addressed my final comments.

I have just noted a problem with the following sentence

"We presume that this is caused by the non-paired data modalities creating a less precise alignment across and in particular within classes since there is a biological connection between data modalities in this synthetic scenario."

I imagine that the authors meant to say that there is NO biological connection between data modalities in this synthetic scenario.

Reviewer #1 (Remarks on code availability):

Code is well maintained

Version 3:

Decision Letter:

Dear Dr Ewald,

We are pleased to inform you that your Resource "AUTOENCODIX: A generalized and versatile framework to train and evaluate autoencoders for biological representation learning and beyond" has now been accepted for publication in Nature Computational Science.

Once your manuscript is typeset, you will receive an email with a link to choose the appropriate publishing options for your paper and our Author Services team will be in touch regarding any additional information that may be required.

Authors may need to take specific actions to achieve compliance with funder and institutional open access mandates.

If your research is supported by a funder that requires immediate open access (e.g. according to <https://www.springernature.com/gp/open-science/plan-s-compliance>) Plan S principles or the <https://www.springernature.com/gp/open-science/us-federal-agency-compliance> NIH public access policy) then you should select the gold OA route, and we will direct you to the compliant route where possible. Because authors warrant under our subscription licensing terms that they haven't committed to licensing any version of their article under a licence inconsistent with the terms of our agreement – including the applicable embargo period – publication under the subscription model isn't suitable for authors whose funders require no embargo.

Acceptance of your manuscript is conditional on all authors' agreement with our publication policies (see <https://www.nature.com/natcomputsci/for-authors>). In particular your manuscript must not be published elsewhere and there must be no announcement of the work to any media outlet until the publication date (the day on which it is uploaded onto our web site).

Before your manuscript is typeset, we will edit the text to ensure it is intelligible to our wide readership and conforms to house style. We look particularly carefully at the titles of all papers to ensure that they are relatively brief and understandable.

Once your manuscript is typeset, you will receive a link to your electronic proof via email with a request to make any corrections within 48 hours. If, when you receive your proof, you cannot meet this deadline, please inform us at rjsproduction@springernature.com immediately.

If you have queries at any point during the production process then please contact the production team at rjsproduction@springernature.com.

We welcome the submission of potential cover material (including a short caption of around 40 words) related to your manuscript; suggestions should be sent to Nature Computational Science as electronic files (the image should be 300 dpi at 210 x 297 mm in either TIFF or JPEG format). We also welcome suggestions for the Hero Image, which appears at the top of our <http://www.nature.com/natcomputsci> home page; these should be 72 dpi at 1400 x 400 pixels in JPEG format. Please note that such pictures should be selected more for their aesthetic appeal than for their scientific content, and that colour images work better than black and white or grayscale images. Please do not try to design a cover with the Nature Computational Science logo etc., and please do not submit composites of images related to your work. I am sure you will understand that we cannot make any promise as to whether any of your suggestions might be selected for the cover of the journal.

Best regards,
Fernando (on behalf of Ananya Rastogi)

--
Fernando Chirigati, PhD
Chief Editor, Nature Computational Science
Nature Portfolio

P.S. Click on the following link if you would like to recommend Nature Computational Science to your librarian: https://www.springernature.com/gp/librarians/recommend-to-your-library

** Visit the Springer Nature Editorial and Publishing website at www.springernature.com/editorial-and-publishing-jobs for more information about our career opportunities. If you have any questions please click here. **

UNIVERSITÄT
LEIPZIG

Universität Leipzig, Data Science Zentrum ScaDS.AI Dresden/Leipzig, Humboldtstraße 25, 04105 Leipzig

Nature Methods
One New York Plaza Suite 4500
New York, NY 10004-1562, USA
methods@us.nature.com

Review response

Dear Dr Ananya Rastogi, dear reviewers

We are thankful for the time and effort of the reviewers in giving us fruitful and constructive feedback to our manuscript. We addressed all raised points which are discussed below in more detail and uploaded an additional version with highlighted track changes. Further, after first manuscript submission, an AUTOENCODIX user discovered an issue in our code where data set splits are used wrongly during embedding evaluation on downstream machine learning tasks (details: <https://github.com/jan-forest/autoencodix/issues/12>). We fixed this issue and re-run all experiments changing and improving previous results substantially but not affecting our main results and insights by benchmarking.

With kind regards,

Maximilian Joas, Neringa Jurenaite, Dusan Prascevic, Nico Scherf, Jan Ewald

ScaDS.AI
DRESDEN LEIPZIG

Center for Scalable Data
Analytics and Artificial
Intelligence

Contact

Dr. Jan Ewald
+49 341 97 39307
Jan.ewald@uni-leipzig.de

28. May 2025

Leipzig University

Data Science Zentrum
Humboldtstraße 25
04105 Leipzig

Web

www.scads.ai

Point-by-point answers

Reviewer #1

The main contribution of this paper is to bring very different approaches using auto-encoders under the same framework and to compare their respective performance in different contexts and under different sets of parameter values. Beyond their ability to produce nonlinear low-dimensional representations of complex datasets, auto-encoders are useful for their generative power and also for their ability to incorporate prior knowledge for interpretability, making them versatile tools with high applicability in multi-omics data analysis. The paper is well written with clear objectives.

The paper provides a very useful broad benchmark between auto-encoder architectures for common tasks such as embedding performance for prognostic clinical models, data modality translation, and explainability by design with ontology-based AEs.

The conclusions that reconstruction capability and embedding quality are not directly related are particularly interesting, and the prescriptions for parameter values are quite useful in practice.

The architectures presented are representative of the state of the art in the field.

Response

We appreciate that the reviewer shares our opinion on the value of our framework, benchmark and choice of architectures.

Main comments:

1) However, I am missing a discussion of the role of paired versus unpaired data. Several recent papers have discussed variations such as vertical, diagonal, or mosaic integration (Argelaguet, Ricard, et al. "Computational principles and challenges in single-cell data integration." *Nature Biotechnology* 39.10 (2021): 1202-1215, He, Zhen, et al. "Mosaic integration and knowledge transfer of single-cell multimodal data with MIDAS." *Nature Biotechnology* (2024): 1-12, Samaran, Jules, Gabriel Peyré, and Laura Cantini. "scConfluence: single-cell diagonal integration with regularized inverse optimal transport on weakly connected features." *Nature Communications* 15.1 (2024): 7762, Cao, Yichuan, et al. "scButterfly: a versatile single-cell cross-modality translation method via dual-aligned variational autoencoders." *Nature Communications* 15.1 (2024): 2973. Tang, Xin, et al. "Explainable multi-task learning for multi-modality biological

data analysis." Nature Communications 14.1 (2023): 2546.). How would this framework manage to handle this variety of pairing settings?

Response

We fully agree with the reviewer that paired vs. unpaired in multi-omics data integration by autoencoders and other methods is an important issue. Since the selected architectures in AUTOENCODIX, except cross-modal VAE, conceptionally require paired data, our framework has currently a focus and limitation on paired data. We added details on this throughout the text in sections Design, Results and Discussion where we include also some of the references thankfully highlighted by the reviewer. Further, the cross-modal with its independent, yet coupled, training shows how unpaired data could be handled in our framework and we outline that we plan to extend our framework to more architectures like masked autoencoders where we can support broader application of AE to scenarios with missing data or unpaired measurements.

2) The code is still quite expert, and the question remains how this kind of approach will be widely adopted by bioinformaticians compared to implementing their own directly. Could you provide simpler tutorials that can be easily run on a laptop? Also, would it be possible to make this framework compatible with the scVerse suite? This would help the adoption rate, as scVerse has recently become the standard in bioinformatics.

Response

We thank the reviewer for the feedback on our tutorials and code. We acknowledge that our current tutorials give too many details for beginners. Hence, we made a new Quick-Start-Guide showing the capability of AUTOENCODIX to train AE with very few lines code and train AE on medium-sized data sets on a laptop without big hardware requirements. We checked runtime on a laptop without GPU acceleration and the X-Modalix (using the *C. elegans* dataset) in the Quick-Start could be trained within 2 mins. Also, other tutorials can be run without GPU on a laptop, e.g. the advanced X-Modalix tutorial using the larger TCGA dataset with MNIST images needed 3-4min for 250+50 epochs.

We share the opinion that compatibility with scVerse is a priority for adoption in the community. We added the functionality that AE embeddings via AUTOENCODIX are stored in the same h5ad-file which is

used for input. In the updated single-cell tutorial we show input/output compatibility with scVerse-tools.

Beyond that, we plan and work on a Python-package version of AUTOENCODIX which will overcome more hurdles and issues in interoperability.

3) Since, as a result of your parameter analysis, reconstruction capability is not directly correlated with performance on downstream tasks in the latent space, one may wonder whether AE is necessarily the best architecture. Can you position your work in comparison to contrastive learning approaches (Richter, Till, et al. "Delineating the effective use of self-supervised learning in single-cell genomics". Nature Machine Intelligence (2024): 1-11.) that create an interesting latent space? Would your framework be easy to extend to this setting?

Response

We agree with the reviewer that the relationship between reconstruction capability and embedding quality is an interesting matter. Related to this, we improved Fig. 2 (using R2, avoided log-scale and rearranged architectures) and, more importantly, the bug-fixed embedding evaluation helped to get more insight into this and other matters. Based on this, partly new, results we highlight now that only Ontix largely improves in both reconstruction and embedding quality. As previously, we are transparent and open about the fact that there is no general superiority of architectures or methods. In particular, we discuss the strong PCA performance (see Reviewer #2 for details) in the text and highlight that disentanglement seems crucial for embedding quality. Related to this, we highlight that the ontology used for Ontix influences or even introduces disentanglement when matching data and downstream task.

Lastly, we added a discussion and outlook to approaches using semi-supervised and contrastive learning approaches which are mentioned by the reviewer. Since they rely on similar architectures and methods as the current set in AUTOENCODIX, we envision with our framework a modular implementation of such ideas in a way that they can be combined easily in the future. In part, this is achievable with our current framework e.g. by adding new architectures, loss functions without the need to implement architectures from scratch. However, for more flexibility we plan a package-version, which would make community contribution even easier.

Minor comments:

Line 111: please provide more details on the latent space density and correlation of the latent dimension. For example, how are these measures normalized? What values would help characterize the quality of an embedding?

Response

We agree that details on these metrics have been sparse and cross-references to the Method section were missing where more details are provided. We added this.

In general, there is no complete set of metrics to assess the quality of embeddings. Nevertheless, our choice reflects the aspects of disentanglement (total correlation), denseness (coverage), retaining high-dimensional data information (reconstruction capability) and, lastly, predictive power for downstream machine learning tasks.

Line 242: Please provide more details on how the training set of images/transcriptomic profile is generated. Are the predicted images time points that are part of the training dataset?

Response

For simplicity and effective training, we split randomly across time points. All shown metrics and predicted images stem from test split samples which were not part of the training set. We added more numbers regarding the split and text for clarity.

Line 464: How did you decide to focus on only one gene? And why this one? Does it have a monotonically changing dynamic over the movie?

Response

We thank the reviewer for pointing out that an explanation is missing. We selected ALY-2 since it provides a strong signal (3rd highest mean) across all time points and, importantly, across cells and substructures of *C. elegans*. The expression patterns can be retrieved and checked directly via the author's website: <https://dulab.genetics.ac.cn/TF-atlas/Gene.html>
We added accordingly text and details.

Reviewer #1 (Remarks on code availability):

The code is hosted on a well documented GitHub with a clear README file and several tutorials as python notebooks designed to help first-time users get familiar with the library.

I have noted however several difficulty when working with MACOS

1) I couldn't find the virtual environment venv-gallia

2) several commands are for linux only like !wget

Please make sure that the tutorial notebooks can be run smoothly from any operating system, not only linux.

Response

We thank the reviewer for feedback on the difficulties with MacOS. We changed the Setup Tutorial and in particular the new Quick-Start-Guide highlights the steps from the ReadMe to do the required steps for each OS. Importantly, different Makefiles are required on different OS. Then the virtual environment can be easily generated via the given make-commands. Finally, we reworked the Tutorials to remove Linux-specific bash-commands. All tutorials run on Linux and MacOS natively.

If problems persist, we are eager to further improve the compatibility on different OS. For that, issue reports by users via Github will be constantly monitored by us.

Reviewer #2

This paper introduces AUTOENCODIX, a unified framework for training, evaluating, and comparing a broad range of autoencoder-based models tailored for biological data. The authors benchmark several models, including Vanillix (standard AE), Varix (β -VAE), Stackix (hierarchical VAE), Ontix (ontology-based VAE), and X-Modalix (cross-modal VAE). The inclusion of Ontix and X-Modalix is particularly innovative and promising for enabling biologically interpretable and modality-bridging representations.

The models are evaluated on both single-cell RNA-seq data and TCGA multi-omics datasets. The authors assess model performance based on two main criteria: (1) reconstruction accuracy and (2) performance on downstream regression and classification tasks using linear models, random forests, and SVMs. Additionally, the authors compare AE-derived embeddings with PCA and UMAP, and use Z-score normalized performance over random feature baselines to enable fair, dimension-aware comparison.

Response

We thank the reviewer for the positive comments about the value of our framework and benchmarks presented in our manuscript.

The main concerns are surrounding the process of comparisons:

On comparison to β VAE. In Fig. 2B, the authors vary the β parameter every 400 epochs to explore disentanglement versus reconstruction trade-offs. However, this experimental design conflates the effects of increased training time and changing β . Since reconstruction typically improves with training time while increasing β penalizes reconstruction through KL divergence, it becomes difficult to isolate the effect of β alone. This should be addressed by either:

- Including fixed- β baselines trained over the same number of epochs
- Visualizing loss trajectories explicitly separated by β rather than epoch count

Response

We agree with the reviewer that the previous experiment was not suitable to completely isolate the effect of β . Hence, we changed the experiment towards the first suggestion and trained individually VAEs for five different β -values over the same number of epochs (1000) as in other experiments. We changed figures, text and tables accordingly. The results are in line with the previous experiment and confirm selection of β -values 0.01, 0.1 and 1 for benchmarks.

On comparison to Stackox (hierarchical AE). The concept of hierarchical stacked autoencoders is well-motivated, but the paper lacks clarity regarding:

- The loss function(s) used for the stacked architecture
- How the model handles different data types, e.g., categorical genotypes vs. continuous expression data
- Whether all modalities share the same latent space prior, and how concatenated latent representations are regularized

Response

We acknowledge that some details on this more complex architecture have been missing and only referencing to the original publication is not ideal. Hence, we added more details, like the prior, in the 'Design and features'-section on stacked VAE and in general how data types (categorical) can be used in AUTOENCODIX. Additionally, we added a remark in the result section on the used MSE-loss for the stacked VAE (similar to all AE).

We hope this now represents a better trade-off between shortness and completeness.

On comparison to Ontix (Ontology-informed AE): Ontix is a conceptually strong model that incorporates domain knowledge (e.g., pathways, chromosomes) into the architecture to provide interpretable latent features. However:

- As shown in Fig. 2C, Ontix consistently underperforms compared to Varix in predictive tasks. The authors should clarify when and why such trade-offs are acceptable (e.g., clinical, regulatory, or exploratory discovery).
- It remains unclear:
 - o Whether the number of latent dimensions corresponds to the number of ontology terms
 - o How overlapping gene-pathway memberships are handled
 - o How many ontology layers are used, and whether one-to-one mappings exist between latent dimensions and biological categories

Despite the lower predictive performance, Ontix's interpretability potential is important. The authors should emphasize case studies or visualizations where Ontix reveals biologically meaningful patterns (e.g., sex-related differences on chromosome X, as briefly mentioned).

Response

Analogously to Stackix description, we added the required details on Ontix. The primary use of Ontix is that latent space dimensions and all decoder nodes are on-to-one mappings to biological pathways or categories. For benchmark completeness we checked performance when introducing additional fully-connected layers to reduce dimension when required dimension (2 or 8) is lower than the number of ontologies (i.e. 29). In the benchmarks shown we use two ontology layers as the sparse decoder. Since ontology relationships are inflicted by masking weights, gene-pathway overlaps are natively supported. Since other implementations name this as source of instability, we have chosen intentionally two ontologies for robustness analysis: with high overlaps (Reactome) and no overlap (chromosomal position). However, we could not confirm the previously named strategy to counter overlap: higher drop-out rates to increase robustness. In our case, the learning rate shows a higher impact on robustness.

The embedding performance of Ontix, after the bug-fix where we unintentionally trained and evaluated the wrong splits (see above), is very good and even slightly better than PCA. Still, we added more discussion of these results and what are primary use-cases for Ontix. In particular, we stress two important points. Firstly, biological-informed VAEs not only introduce explainability, but can even improve results since fewer

parameters need to be trained, avoiding overfitting. Secondly, disentanglement of embedding dimensions can be introduced via the usage of an appropriate ontology matching the downstream prediction task. The latter is connected in the text now with the chromosomal embedding example where the signal on the X-chromosome is very strong. Since our manuscript already is quite long and complex, we refrained from adding additional visualizations, experiments and analyses at this point. Nevertheless, we hope to have convinced readers on the potential of ontology-based VAEs by the shown examples and insights.

On the comparison to X-Modalix (Cross-modal AE): The motivation behind X-Modalix — to enable translation across omics modalities — is compelling. However, the main demonstration (gene expression → MNIST digits) is synthetic and biologically unmotivated.

Given that TCGA contains real multi-modal data (expression, methylation, miRNA, etc.), it is unclear why a real modality-to-modality translation (e.g., expression ↔ methylation) was not used to evaluate the model's utility. At minimum, the authors should justify this choice and address whether technical limitations (e.g., data sparsity, alignment) prevented a more biologically relevant experiment.

Response

We understand that the synthetic combination of TCGA and MNIST does not directly provide biological relevance. Hence, we included already in the first submission the second experiment and dataset of *C. elegans* development. We added additional justification now in the text. The main reason is that a translation of Omics-to-image can be checked easily visually. In particular, MNIST digits are directly checkable if they match the cancer class and the assigned digit. Based on this, we can learn better how to set-up the complicated loss function weightings to ensure both precision and latent space alignment. This qualitative and quantitative evaluation is barely possible for Omics-to-Omics translation where references, experimental validation and gold-standards are missing to judge "good" translations.

To address the gap that we do not provide Omics-to-Omics translation, we performed an additional experiment where we train a X-Modalix for the same cancer subtypes, but translating DNA Methylation to gene expression. Judging that the gene expression is successfully translated, is difficult to evaluate. However, we show in Supplemental Material that overall structure and information is retained when comparing UMAPs of

the original gene expression with the reconstructed and translated one. Additionally, we speculated if embedding quality may even increase by latent space alignment and its regularizing pressure on each VAE training. Hence, we checked the embedding quality on downstream tasks in comparison to normal trained VAE but found no improvement or decay. We make cross-references on these results in the main text and stressed the point that future work in Omics-to-Omics translation will require experimental validation to evaluate translation capabilities towards a single-gene level.

Here are additional concerns:

1. Missing R^2 for AE Reconstruction:

While the authors use R^2 for CNN regression (Table S4), reconstruction quality for autoencoders is reported only via raw loss, which is scale- and sample-size dependent. R^2 is a scale-invariant and interpretable measure of variance explained. Reporting R^2 for each AE variant would provide much-needed clarity in comparing reconstruction effectiveness.

Response

We fully agree with the reviewer that showing absolute reconstruction loss was not ideal to gain comprehensive insights. We changed all related figures and text showing now the explained variance R^2 to assess reconstruction capability. Shown values are averages over all (test) samples.

2. Strong PCA Baseline (Fig. 2D):

PCA with 29 components performs comparably — or better — than many AE models. Given its simplicity and interpretability, the authors should explicitly discuss when PCA is insufficient and whether AE models truly offer added value for downstream biological tasks.

Response

Despite the new results, PCA remains a strong baseline. As suggested by the reviewer, we added more discussion and detail from a user-perspective in the discussion section. We shared here our view based on the results that UMAP like Stackix are very strong at producing very low-dimensional representation (2D). However, PCA like Ontix are more valuable to obtain disentangled and meaningful representations which overall improve predictive power of embeddings. Notably, in new results

Ontix with 29-dimensional Reactome-inspired latent space shows slightly better results as PCA. Further, we mention now an additional point in the text. We make the point that Ontix via the choice of ontology gives the opportunity to focus on other sources of variance which are not the primary source as shown in the X-chromosome example in Fig. 3. Here, a strong male/female signal is obtained, but primary source of variance is tissue in TCGA.

3. Latent Dimension vs. Classifier Interaction (Fig. S2):

The paper reports that larger latent spaces improve performance for linear models, but degrade performance for SVMs and RFs. This trend deserves deeper analysis:

- Is this due to noise accumulation, feature redundancy, or curse of dimensionality?
- Would feature selection or sparse priors mitigate this?

Understanding this interaction is critical for practical deployment of latent embeddings.

Response

This phenomenon is in the new results, based on the correctly used test splits during embedding evaluation, not as apparent as before.

We, and others, judge embedding quality mainly on linear models since classes should be ideally linear separable in the latent space. For completeness we checked RandomForest and SVM (radial-kernel matching Gaussian VAEs). However, the hyperparameters of these methods are not optimized and we see varying under- or overfitting problems throughout the machine learnings tasks. To assess the full potential of SVM and RF, tuning each task and architecture combination would be required, vastly increasing runtime of our benchmark.

Additionally, our baseline performance, randomly chosen features, also increases with latent dimension since by chance predictive features are chosen. In some cases and machine learning tasks, non-linear methods RF and SVM show stronger performance increase on a high number of randomly chosen features in comparison to linear models.

Summary:

AUTOENCODIX is a potentially valuable contribution that brings a unified, modular framework to biological representation learning. It covers a wide landscape of model designs and offers a useful experimental protocol. However, several

evaluation design issues, including incomplete metric reporting, overreliance on synthetic data, and insufficient biological justification for some models, weaken the paper's clarity and impact.

With clarification, more complete biological evaluation, and consistent R^2 -based metrics, this framework might serve as a benchmark platform for future bio-AE research.

Response

We thank the reviewer again for the fruitful and constructive feedback. We addressed all points raised by the reviewers and could improve our manuscript and framework.

Reviewer #2 (Remarks on code availability):

Code is well maintained.

Response

We appreciate the positive review of our code basis.

UNIVERSITÄT
LEIPZIG

Universität Leipzig, Data Science Zentrum ScaDS.AI Dresden/Leipzig, Humboldtstraße 25, 04105 Leipzig

Nature Methods
One New York Plaza Suite 4500
New York, NY 10004-1562, USA
methods@us.nature.com

Review response

Dear Dr Ananya Rastogi, dear reviewers,

We thank the reviewers again for their time and effort during the second revision round. We addressed the raised points which are discussed below in more detail and uploaded an additional version with highlighted track changes. In particular, we hope that we have addressed now all issues on MacOS.

With kind regards,

Maximilian Joas, Neringa Jurenaite, Dusan Prascevic, Nico Scherf, Jan Ewald

ScaDS.AI
DRESDEN LEIPZIG

Center for Scalable Data
Analytics and Artificial
Intelligence

Contact

[Dr. Jan Ewald](mailto:Jan.Ewald@uni-leipzig.de)
[+49 341 97 39307](tel:+493419739307)
Jan.ewald@uni-leipzig.de

7. July 2025

Leipzig University

Data Science Zentrum
Humboldtstraße 25
04105 Leipzig

Web

www.scads.ai

Point-by-point answers

Reviewer #1

The authors have answered most of my comments. They mention a bug in the implementation that has been identified between the previous and current version of the article, however the main messages are not affected.

Response

We thank the reviewer for his overall positive review of our revised manuscript.

I still have an important remark

The authors have clarified that their architectures are meant to be used on paired data. In particular they write in the introduction "Currently, our implemented architectures rely on paired measurements which are often difficult to obtain in multi-omics settings". However, in the result section corresponding to cross-modal VAE X-modalix, the model is evaluated on unpaired data (only at the class level). They write "Our scenarios cover two cases either having non-paired data modalities .. and paired measurements. "Notably, although our framework is not designed to handle unpaired multi-omics data, the first scenario shows that for X-modalix we can support this by using the class-based loss." It is unclear how the artificial pairing is created, is it a random shuffle of samples of similar classes that are paired randomly? In that case, how is the shuffling controlled and is it possible that it would bias the data? I am wondering about the relevance of the TCGA to MNIST experiments. The relationship between the two modalities is highly artificial as there is in principle no mechanism linking RNASeq to written digits. This type of example for illustrating the translation ability of cross-modal autoencoders might even induce suspicion on the biological relevance of this task. Previous authors have warned that non-linear methods such as UMAP can hinder biological relevance, e.g. "The specious art of single-cell genomics" <https://journals.plos.org/ploscompbiol/article?id=10.1371/journal.pcbi.1011288> . I am slightly worried that the example, RNASeq to MNIST translation proposed by the authors might lead to similar worry. Since the TCGA RNASeq to methylation use case works well (and is performed on paired data), I am wondering if it wouldn't be more relevant to include in the main text instead of the RNASeq to MNIST use case.

Similarly, I do not understand the statement that "We intentionally focus on Omics-to-image scenarios because meaningful translation can be visually checked

whereas Omics-to-Omics translation typically needs experimental validation to prove biological correctness". I do not agree that looking at realistic images is sufficient to assess validity of the approach, especially when the link between the RNASeq to image is largely artificial, and if anything, shows the ability of the method of performing clustering. Label being associated to the average image of the class. In the Yang et al. paper that the authors use as a starting point, the assumption that the various modalities are governed by the same underlying latent variable is clearly stated and should be a prerequisite for this type of approaches.

Response

We understand the reviewer's concerns on this synthetic and artificial linking of data modalities. We took the advice and moved the real-world scenario of DNA methylation to RNASeq gene expression into the figure of the main text. We adjusted the text and figures to make the point that the TCGA-MNIST scenario is meant as a controlled synthetic scenario to explore loss-term weightings controlling latent space alignment and serving as scenario where alignment relies purely on class information rather than paired multi-omics measurements.

We also clarified now in the text (result and method description) that digits from the MNIST dataset were selected randomly from the respective digit class which we artificially assigned to the cancer type. Hence, we intentionally did not introduce any further bias or link between the data modalities allowing us to prove that this weak artificial link is sufficient to enable translation between modalities.

Lastly, we experienced that visually checking images from translation, when we tried different loss term weightings and hyperparameter combinations, was very helpful and easier than verifying plausibility of gene expression data vectors.

(Remarks on code availability):

The code is well maintained. I tried to run the quick start tutorial https://github.com/jan-forest/autoencodix/blob/main/Tutorials/Quick_Start.ipynb - however several limitations appeared with Makefile - due probably to the way indentation are encoded in the text editor used by the authors. Please double check on macOS.

Response

We are sorry that there was still an issue to use our code and Quick-Start Tutorial on MacOS. We tried on several configurations and luckily could

identify that it was a unique issue when using Jupyter-Lab and MacOS. Since we have been unhappy with the necessity of different Makefiles for MacOS and Linux, we rewrote substantial parts and now have a uniform Makefile for both OS. We thank the reviewer for pointing out this issue.

Reviewer #2

(Remarks to the Author):

All the comments has been addressed, I have no further comments.

Reviewer #2 (Remarks on code availability):

The code are good to use.

Response

We are happy to read that we could address all concerns and we thank the reviewer again for his/her valuable comments in the first round.